# Feedback-Guided Black-box Attack in Federated Learning: A Cautious Attacker Perspective

## Abstract

Federated Learning (FL) is a robust approach to collaborative machine learning that upholds the integrity of data privacy by ensuring that data remains with the owners. However, FL systems are vulnerable to sophisticated adversarial attacks from malicious clients, especially those leveraging black-box settings. Unlike centralized data poisoning, attacking FL presents unique challenges (i) server-side defense mechanisms can detect and discard suspicious client updates, requiring attacks to maintain minimal visibility across multiple training rounds, and (ii) malicious clients must repeatedly generate poisoned data using only their local black-box model for each round of training, as previous poisoning attempts may be nullified during global aggregation. This forces adversaries to craft stealthy poisoned data locally in a black-box context for each round, maintaining low visibility while ensuring impact. Existing FL attack methods often show high visibility while maintaining impact due to their attack nature, the scale of the introduced perturbations, and the lack of detection strategies. Also, these methods often rely on maximizing cross-entropy loss on the true class, resulting in delayed attack convergence and highly noticeable perturbations. Hence, it is crucial to develop a stealthy data poisoning attack with low visibility for black-box settings in order to comprehend the motives of a cautious attacker in designing an FL attack. To address these challenges, we propose a Feedback-guided Causative Image Black-box Attack (F-CimBA), which is specifically designed for FL by adding random perturbation noise to the data. F-CimBA minimizes the loss of the most confused class (i.e., the incorrect class that the model confuses with the highest probability) instead of the true class, allowing it to exploit local model vulnerabilities for early attack convergence. This approach ensures that poisoned updates maintain low visibility, reducing the likelihood of server-side rejection. Furthermore, F-CimBA adapts effectively under non-IID data distributions and varying attack scenarios, consistently degrading the global model's performance. Additionally, we analyze its impact on system hardware metrics, highlighting the stealth and efficiency of F-CimBA, considering the computational overhead of repeated poisoning attempts in the FL context. Our evaluation demonstrates F-CimBA's consistent ability to poison the global model with minimal visibility under varying attack scenarios and non-IID data distributions, even in the presence of robust server-side defenses.

## 1 Introduction

Federated learning (FL) has emerged as a powerful machine learning paradigm because of its use in several applications, such as healthcare, signal processing, mobile user personalization, computer vision, etc. (Wang et al., 2023; Zeng et al., 2023). Despite the advantages of shared intelligence, the decentralized nature of FL makes it highly susceptible to adversarial attacks (Chen et al., 2022; Sun et al., 2023; Lewis et al., 2023). These attacks are known as poisoning attacks (Kumar et al., 2024a;b) and are further classified into (i) *Data poisoning,* where compromised clients only provide malicious data without interfering with the local model training (Shejwalkar et al., 2022). (ii) *Model poisoning,* where, compromised clients directly provide malicious updates (Baruch et al., 2019; Fang et al., 2020). In this paper, we focus on online untargeted black-box data poisoning attack as it is the most common and relevant to real-world FL deployments, as

outlined in the recent work by (Shejwalkar et al., 2022). In the context of black-box attacks, the key distinction between federated and centralized learning lies in the attacker's access to the global model's internal information. In centralized learning, attackers manipulate input data or query the model without access to its parameters, while in federated learning, attackers participate as malicious clients, indirectly influencing the global model through manipulated local updates. Although the core concept remains similar, attack strategies and implications differ due to the distinct nature of these learning systems (Kumar et al., 2024b). Also, these black-box data poisoning attacks remain undetected for a longer duration of time among a larger population of FL clients. Specifically, our attacker is interested in generic misclassification (untargeted) rather than specific misclassification (targeted).

**Limitations of existing attacks.** Table 1 provides a summary and gaps in existing FL attacks, highlighting their properties, limitations, and broad scope of our proposed method. Existing recent works such as PGA (Shejwalkar et al., 2022), ADA (Sun et al., 2022), internal evasion, (Kim et al., 2023), etc, have exposed FL's high vulnerability to minor perturbations of adversarial attacks under the white-box setting. Executing a white-box attack in real-world scenarios becomes impractical, as it necessitates knowledge of internal model details, which is restricted due to multiple layers of model security. On the other hand, in a black-box adversarial attack, the adversary can create malicious data with minimal information about the model and subsequently train the local model on the poisoned data to generate a malicious update. After a few rounds of updates, the malicious attack in FL forces the global model at the server model to cause drastic misclassification with extremely high confidence (Shejwalkar et al., 2022). Moreover, the focus on black-box data poisoning attacks has predominantly been within centralized machine learning settings. There is a scarcity of research on untargeted black-box causative data poisoning attacks specifically tailored for FL environments, as explored in prior work (Mothukuri et al., 2021). Furthermore, malicious actors deliberately insert poisoned data into the client training datasets to induce incorrect predictions while evading detection, resulting in low attack visibility. This is essential for ensuring covert execution, enabling attackers to achieve their goals without activating the server's detection mechanisms. Typically, attackers strive to maximize the impact of their attacks while maintaining minimal visibility. Therefore, considering these two factors is vital for understanding the adversary's approach to designing attacks in real-world scenarios. To address these challenges, we propose a robust poisoning attack strategy that achieves low visibility and high impact in causative black-box data poisoning scenarios.

Table 1: Comparison of F-CimBA with existing causative attacks across key dimensions. For each desired property, we indicate whether it is explicitly considered and optimized (✔) or not explicitly addressed (✗) in the design of various methods. C-client, S-server, DP-data poisoning, and MP-model poisoning.

| Attack Method | Attack Type | Attack Setting | Attack Impact | Attack Visibility (Stealthiness) | Different non-IID Data Analysis | Attack Impact Analysis on Hardware Metrics |
|---|---|---|---|---|---|---|
| DPA-DLF (Shejwalkar et al., 2022) | DP | B | ✔ | ✗ | ✗ | ✗ |
| DPA-SLF (Shejwalkar et al., 2022) | DP | B | ✔ | ✗ | ✗ | ✗ |
| PGA (Shejwalkar et al., 2022) | MP | W | ✔ | ✗ | ✗ | ✗ |
| LIE (Baruch et al., 2019) | MP | W | ✔ | ✔ | ✗ | ✗ |
| DYN-OPT (Shejwalkar & Houmansadr, 2021) | MP | W | ✔ | ✗ | ✗ | ✗ |
| ADA (Sun et al., 2022) | MP | W | ✔ | ✗ | ✗ | ✗ |
| Edge-case backdoor (Wang et al., 2020) | DP, MP | W | ✔ | ✗ | ✗ | ✗ |
| Stealthy model poison (Bhagoji et al., 2019) | MP | W | ✔ | ✔ | ✗ | ✗ |
| (Bagdasaryan et al., 2020) | MP | W | ✔ | ✗ | ✗ | ✗ |
| (Fang et al., 2020) | MP | W | ✔ | ✔ | ✔ | ✗ |
| F-CimBA (ours), 2024 | DP | B | ✔ | ✔ | ✔ | ✔ |

In this paper, we introduce a causative black-box data poisoning adversarial attack on FL called **feedback-guided causative image black-box adversarial attack (F-CimBA)** using gradient based perturbation. The proposed attack minimizes the loss of the most confused class, i.e., the target class that the model confuses with the highest probability. In an untargeted attack, boosting the probability score of the most confused class provides advantages like increased misclassification rates, leveraging model weaknesses, and faster attack convergence. Additionally, our approach utilizes the probability score of the most confused class as feedback to iteratively refine adversarial perturbations that remain almost imperceptible to human

observers. We include two key attack settings in our threat model such that the attacker controls the data used in a single or multi-client attack. Further, the attacker has control of all the compromised clients' data and only their model predictions. Figure 1 shows the F-CimBA method on the client side. The poisoned local model update tries to disrupt the benign global model parameter values (red) by reducing its variance and dropping the test accuracy. We conduct extensive experiments using homogeneous and heterogeneous data shards with different numbers of clients. We present time-series information on scalability for attack metrics such as misprediction accuracy, global epochs for model convergence, and its effect on hardware performance metrics like central processing unit (CPU) and graphics processing unit (GPU) utilization. Furthermore, we conduct an extensive evaluation of F-CimBA's robustness in the presence of Byzantine-robust aggregation techniques such as Krum (Blanchard et al., 2017), trimmed mean (Yin et al., 2018), and median (Yin et al., 2018) at the server level. Also, we assess the performance of our method under the recent robust defense mechanisms like FLTrust (Cao et al., 2021), local malicious factor (LoMar) (Li et al., 2023), and FLDefender (Jebreel & Domingo-Ferrer, 2023) to comprehensively measure its significance and effectiveness.

Below are the main contributions of our work to the field of adversarial attacks in FL systems.

- We present a novel causative black-box data poisoning adversarial attack in federated learning, employing gradient-based perturbation from the perspective of a cautious attacker. Our approach involves crafting a stealthy attack, strategically balancing high attack impact with low attack visibility factors. Also, we introduce a feedback mechanism that utilizes the probability of the most confused class to update the adversarial perturbation towards achieving low attack visibility with high attack impact.

- We provide theoretical proof for the convergence, computational, and communication costs of our proposed method.

- We perform a comprehensive analysis to assess the potential implications of the proposed F-CimBA method for degrading the performance of system hardware performance metrics. These metrics encompass crucial parameters such as execution time, bandwidth, CPU percentage, GPU memory, GPU utilization, and GPU temperature. Also, we analyze the impact of non-independent and non-identically distributed (non-IID) data on our F-CimBA attack.

- We conduct a comprehensive analysis through extensive experiments involving four diverse datasets and three different models. Specifically, we utilize a convolutional neural network (CNN) model for the German traffic sign recognition benchmark (GTSRB) (Stallkamp et al., 2011) and KUL Belgium traffic sign dataset (KBTS) (Mathias et al., 2013) datasets. Also, we use a residual network (ResNet18) for the Canadian institute for advanced research (CIFAR10) (Krizhevsky et al., 2009) dataset and the LeNet architecture for the extended modified national institute of standards and technology (EMNIST) (Cohen et al., 2017) dataset. Furthermore, our evaluation of F-CimBA extends across both homogeneous and heterogeneous FL data settings, exploring scenarios with varying total numbers of clients and attack percentages.

**Paper outline:** Section 2 presents related work of existing FL data poisoning and model poisoning attacks. Section 3 provides essential background information and threat model details. Section 4 provides the details of the F-CimBA algorithm. Section 5 presents the convergence analysis of F-CimBA. The computational and communication cost is given in Section 6. Section 7 presents a comprehensive assessment of F-CimBA, covering different FL settings, attack scenarios, system configurations, and machine settings. Section 8 provides a detailed ablation study of our work. Section 9 discusses the paper's limitations and outlines future work. Finally, Section 10 concludes with key research findings and future research directions.

## 2 Related Work

This section provides a comprehensive overview of existing attacks in FL. As discussed in Section 1, adversarial attacks in FL are generally classified into two types: data poisoning attacks and model poisoning attacks.

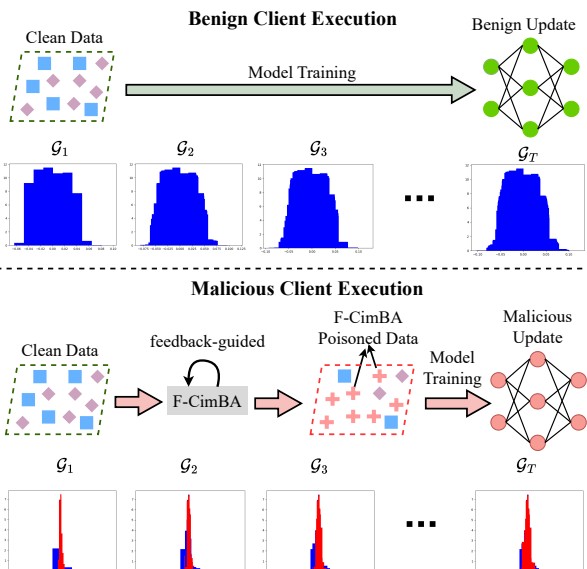

Figure 1: Attack overview: malicious clients poison training data by inducing feedback-guided perturbations to generate malicious updates. We show the attack effect on the global model ($\mathcal{G}$) weights for $T$ communication rounds. Eventually, F-CimBA malicious update (red) reduces the variance of the benign global model weights (blue) and thereby drops the test accuracy.

**Data Poisoning Attacks:** Adversarial attacks on machine learning models and deep neural networks have gained significant attention (Vadillo et al., 2023; Rakhsha et al., 2021). While most research has focused on centralized ML (Yang et al., 2020), there has been limited exploration of untargeted black-box data poisoning attacks in FL environments. For instance, (Bagdasaryan et al., 2020) introduced a backdoor attack framework for FL, leveraging a constrain-and-scale technique to train on backdoor data and submit a compromised model update to the central server. Similarly, (Fang et al., 2020) treated labelflip attacks as optimization problems, applying them to Byzantine-robust FL methods. (Shejwalkar et al., 2022) proposed two variations of data poisoning attacks, static labelflip (DP-SLF) and dynamic labelflip (DP-DLF), to demonstrate the vulnerabilities in FL systems. Each of these approaches underscores specific risks in FL. For example, Bagdasaryan *et al.*'s(Bagdasaryan et al., 2020) method highlights the dangers of backdoor attacks, emphasizing the importance of robust countermeasures. (Zhang et al., 2020) showcased generative adversarial attacks' effectiveness in poisoning FL. Fang *et al.*(Fang et al., 2020) exposed the susceptibility of FL to label manipulation, aiding the development of Byzantine-robust techniques. Lastly, (Shejwalkar et al., 2022)'s work illustrates the threats posed by poisoned training data, stressing the critical need for efficient detection and defense mechanisms.

**Model Poisoning Attacks:** This category involves attackers directly submitting malicious updates (Bhagoji et al., 2019; 2018). Various strategies have been proposed to craft such updates effectively. For example, (Baruch et al., 2019) introduced the "little is enough" (LIE) attack, where noise is added to the average of benign updates using the standard deviation of these updates to generate poisoned updates. (Shejwalkar & Houmansadr, 2021) designed malicious model updates by perturbing the benign reference aggregate in a direction that maximally aligns with the attacker's intent. Similarly, (Fang et al., 2020) computed the average of benign updates, identified a fixed malicious direction, and generated poisoned updates by determining suboptimal parameters to bypass the aggregation rule. Each of these approaches reveals critical vulnerabilities in federated learning systems. Baruch *et al.*'s LIE attack demonstrates the impact of subtle noise injection into the aggregation process. Shejwalkar *et al.*'s technique highlights the ability to manipulate learning dynamics through targeted perturbations of the reference aggregate. Fang *et al.*'s strategy emphasizes how carefully crafted updates can evade aggregation rules and degrade the quality of the federated model.

Existing attack methods have several limitations, such as insufficient analysis of FL settings (both homogeneous and heterogeneous), a lack of realistic attack scenarios as highlighted by (Shejwalkar et al., 2022), and the computational overhead associated with GAN-based methods. Moreover, black-box adversarial attacks in FL using gradient perturbation remain largely unexplored. Our proposed approach addresses these limitations through a feedback-guided causative black box attack method in an FL setup with lower visibility. Additionally, we analyze system hardware performance metrics after the attack, a direction not previously explored by other works.

## 3  Preliminaries

This section provides the essential background information required to comprehend the remainder of the paper. A summary of the notations used throughout this work is presented in Table 2.

Table 2: Overview of notations used.

| Notation | Definition |
|---|---|
| $n$ | Total number of clients |
| $k$ | Client's index |
| $m$ | Total number of malicious clients |
| $T$ | Global epochs |
| $E$ | Local epochs |
| $\mathcal{D}_k$ | $k^{th}$ client local data |
| $\tilde{\mathcal{D}}_k$ | F-CimBA poisoned $k^{th}$ client local data |
| $\mathcal{G}_{\theta_g}$ | Global model with parameters $\theta_g$ |
| $f_\theta$ | Local model at epoch $\theta$ |
| $\mathcal{D}_{test}$ | Test data at the server |
| $\eta$ | Learning rate |
| $\mu$ | Noise coefficient |
| $B$ | Batch size |
| $B_n$ | Total batches |
| $\mathcal{L}$ | Cross-entropy loss function |
| $\mathcal{A}$ | Attacker client |
| $\lambda_i$ | Weighted coefficient of FedAvg function |
| $\mathcal{U}$ | Random gradient perturbation noise |

**Definition 3.1** *(**FL setup.**) We consider a FL system consisting of a central server and n clients. Each client, denoted as $\mathcal{C}_k$, where $k \in [1, n]$, has its private local dataset $\mathcal{D}_k$, referred to as a shard. Specifically, $\mathcal{D}_k = \{(\mathcal{X}_{i,k}, \mathcal{Y}_{i,k})\}_{i=1}^{\mathcal{N}_k} \subseteq \mathbb{R}^d \times \mathbb{R}$, where $\mathcal{X}_{i,k}$ represents the input data and $\mathcal{Y}_{i,k}$ denotes the corresponding labels. For simplicity, we assume that $\|\mathcal{X}_{i,k}\|_2 = 1$ for all $i \in [1, \mathcal{N}_k]$ and $k \in [1, n]$, with the final component of each feature vector fixed at $\mathcal{X}^d = 1/2$, ensuring $L_2$-norm data normalization. The primary objective of FL is to collaboratively train a global model $\mathcal{G}_{\theta_g}$ (where $\mathcal{G}$ represents the model and $\theta_g$ denotes its parameters) that generalizes well on a global test dataset $\mathcal{D}_{test}$. During each training round t, the central server distributes the current global model parameters $\theta_g^t$ to all n clients. Upon receiving $\theta_g^t$, each client $\mathcal{C}_k$ initializes its local model parameters $\theta_k^t$ with $\theta_g^t$ and trains the local model using its private dataset $\mathcal{D}_k$ with a batch size B and E local epochs. After local training, the updated local parameters are computed as: $\theta_k^t = \theta_k^t - \eta \nabla \mathcal{L}(\mathcal{X}_k, \mathcal{Y}_k)$, where $\eta$ is the learning rate and $\mathcal{L}$ is the local cross-entropy loss. Each client then sends its local model parameters $\{\theta_k^t\}_{k=1}^n$ back to the server for aggregation. The central server aggregates these local models using the Federated Averaging (FedAvg) method (McMahan et al., 2017), which can be expressed as:*

$$\theta_g^{t+1} = \sum_{k=1}^n \lambda_k \theta_k^t, \tag{1}$$

*where $\lambda_k = \frac{\mathcal{N}_k}{\sum \mathcal{N}_k}$ and $\sum_k \lambda_k = 1$. This iterative process is repeated until the global model achieves convergence, satisfying the following condition $\arg\min_{\theta_g}[\mathcal{L}(\theta_g) \triangleq \frac{1}{n} \sum_{k=1}^n \mathcal{L}(\theta_g, \mathcal{D}_k)]$.*

Table 3: Key dimensions of our threat model and their attributes.

| Objective | | | Knowledge & Capabilities | | Attack Mode |
|---|---|---|---|---|---|
| Security violation | Attack specificity | Error specificity | Model | Data distribution | Consciously active |
| **Availability:** Misclassify test data and cause disruption to benign clients' objectives. | **Indiscriminate:** Misclassify all or most of the test inputs during inference. | **Untargeted:** Misclassify the give test data to any other class. | **Black-box:** Adversary cannot break into the compromised clients and cannot manipulate the model parameters. | The adversary can only access the local data distributed at the clients. **Note: The attack is agnostic to the type and degree of non-IID in the distributed data at the clients.** | **Online:** The adversary repeatedly and adaptively poisons the model based on the attack budget strategy. |

We adopt an FL data shard setting with non-independent and non-identically distributed (non-IID) data. This is achieved by partitioning the training data using a Dirichlet distribution (Minka, 2000) with parameter $\beta$ among clients. This sharding approach allows us to adeptly manage data heterogeneity, mirroring the nature of the real-world data distribution. Additional information on Dirichlet data distribution are provided in Section 7.

### 3.1 Feedback-guided Gradient Perturbation for Adversarial Sample Generation

Optimization typically requires guidance to find gradients that minimize the cross-entropy loss on true class. In our case, we need to minimize the loss of the 'most confused class' (as defined in Section 1) by searching for the gradient perturbations. Earlier works like Gressel *et al.* (Gressel et al., 2021) proposed a feature importance guided attack (FIGA) that perturbs the most important features of the input towards the target class. We extend this idea to propose a feedback-guided attack that searches for gradient perturbations to minimize the loss of the most confused class.

### 3.2 Threat Model

In this paper, we adopt a threat model grounded in real-world FL production environments, where malicious clients inject poisoned data and train their local models using this compromised dataset. These clients operate under black-box constraints, meaning they cannot manipulate the training process, alter the server's aggregation algorithm, or interfere with server communications. However, they retain control over their local data and have the ability to access predictions generated by their local model on any selected input. Our threat model aligns with the framework proposed by (Shejwalkar et al., 2022), recognized as one of the most realistic and practical approaches for FL scenarios. Real-world scenarios relevant to our threat model include (i) healthcare applications where FL may be employed to train models on medical data from various hospitals. Malicious participants can introduce corrupted data to compromise the model's performance. (ii) In FL-based autonomous driving applications, compromised vehicles can provide manipulated data to the central server, affecting the overall model's performance. (iii) Fintech applications domain where financial institutions can collaborate using FL to detect fraudulent transactions. In such a scenario, a malicious client can poison the data to evade detection or reduce the overall model's performance. Based on this threat model, we present the critical dimensions of our threat model and the assumptions we make about the FL setup as shown in Table 3. The attacker's primary goal is to cause the global model, used for testing on the server, to misclassify most or all of the test data, thereby reducing its performance. This attack is untargeted, aiming for generic misclassification rather than specific, targeted misclassifications. The attacker operates under a black-box assumption regarding the server, with no access to the server's parameters, global model predictions, or aggregation process, and assumes the server is trustworthy and does not inspect model updates. On the client side, the attacker controls the data of one or more compromised clients, which generate updates based on trusted local training processes. However, the attacker cannot interfere with the training procedures or communication between the compromised clients and the server, being limited to manipulating the clients' local data without affecting their training or server interaction. Further, we consider an active attacker employing repeat and adaptive data poisoning strategies on the compromised clients. These strategies utilize feedback from F-CimBA to sustain the attack throughout the FL process (online attack).

## 4    Proposed Approach

This section describes the details of our F-CimBA configured FL system.

**Problem statement.** In each round $t$, the server sends the global model parameters $\theta_g^t$ to all $n$ clients, out of which $m$ clients are malicious *s.t.* $m > 0$ & $m << n$. The adversarial client $\tilde{\mathcal{C}}_k$ introduces a $\mu$-bounded poisoned data followed by one-pixel mutations into training dataset $\tilde{\mathcal{D}}_k$ such that $\tilde{\mathcal{D}}_k = \{\tilde{\mathcal{X}}_i, \mathcal{Y}_i\}_{i=1}^{\mathcal{N}_k}$, where $\mathcal{N}_k$ is the training data size, as defined in Definition 3.1.

**Definition 4.1** ($\mu$-**bounded adversary.**) *Let $\mathcal{F}$ be the adversary's function class. An adversarial perturbation $\mathcal{U}$ is characterized by the mapping $\mathcal{U} := \mathcal{F} \times \mathcal{X} \times \mathbb{R} \rightarrow \tilde{\mathcal{X}}$. For $\mu > 0$, we define the $L_2$ norm ball as $\mathcal{B}_2(\mathcal{X}, \mu) := \{\tilde{\mathcal{X}} \in \mathbb{R}^d : \|\tilde{\mathcal{X}} - \mathcal{X}\| \leq \mu\} \bigcap \mathcal{X}$. We classify the adversary $\mathcal{A}$ as $\mu$-bounded if it adheres to the condition $\mathcal{U}(\mathcal{F}, \mathcal{X}_i, \mathcal{Y}_i)_{i=1}^{\mathcal{N}_k} \in \mathcal{B}(\mathcal{X}_i, \mu)_{i=1}^{\mathcal{N}_k}$. Furthermore, for a given $\mu > 0$, we denote the worst-case adversarial perturbation $(\mathcal{U}^+)$ as*

$$\mathcal{U}^+ := \underset{\tilde{\mathcal{X}} \in \mathcal{B}(\mathcal{X}, \mu)}{\arg\max} \mathcal{L}(f_\theta(\tilde{\mathcal{X}}_i, \mathcal{Y}_i)_{i=1}^{\mathcal{N}_k}),$$

*where $\mathcal{L}$ and $f_\theta$ represent the cross-entropy loss function and parameters of the local black-box model, respectively.*

The adversarial client $\tilde{\mathcal{C}}_k$ trains the local model using the poisoned data and sends malicious parameters $\tilde{\theta}_k^t$ to the server. The server aggregates the parameters sent by malicious and benign clients as per Eq. (1) in Definition 3.1 and performs testing. Let $\mathcal{A}_\mathcal{G}$ and $\mathcal{A}_\mathcal{G}^*$ be the global test accuracy achieved by the global model $\mathcal{G}_{\theta_g}$ before and after the attack, respectively. The problem statement from the attacker's perspective involves generating adversarial perturbed training data $\tilde{\mathcal{D}}_k$ with the following objectives:

1. Maximize the impact of the attack with a lower global test accuracy such that $\mathcal{A}_\mathcal{G}^* << \mathcal{A}_\mathcal{G}$.

2. Maximize the attack impact while ensuring minimum attack visibility $V$ denoted as $V = |Val_\mathcal{G} - Val_{\tilde{\mathcal{C}}}|$, where $Val_\mathcal{G}$ and $Val_{\tilde{\mathcal{C}}}$ are the validation accuracy of global model and the local adversarial client model, respectively.

We outline the changes to the standard FL framework, both on the server and client sides, to incorporate the F-CimBA attack. A comprehensive flowchart of our method is shown in Figure 2. On the server side, no modifications are introduced since our focus is primarily on the data poisoning attacks executed at the client level. In each training epoch, the malicious client $\mathcal{C}_k$ receives the global model parameters $\theta_g^t$ from the server. These parameters are then used to train the local model $\theta_k^t$ on the adversarially poisoned data. This process enables the computation of the malicious model update $\hat{\theta}_k^t$.

**Attack motivation.** As detailed in Section 1, our research objective is to achieve maximum attack impact with minimal attack visibility under black-box attack settings. Limited prior work has studied untargeted black-box causative data poisoning attacks applicable to FL settings. Further, they often lack alignment with the attacker's perspective in generating a high-impact attack with low visibility. Therefore, the demand arises for a relatively straightforward yet effective poisoning attack that drastically affects the accuracy while evading detection (low visibility).

**Proposed F-CimBA attack.** To design a black-box online data poisoning attack, we propose a novel feedback method for our F-CimBA attack. Here, we minimize the loss of the most confused class. It is the incorrect class where the model misclassifies with the highest probability. Raising the most confused class probability score in an untargeted attack leads to (i) higher misclassification rates by raising the ratio of misclassified samples and enhancing the confusion matrix. (ii) Faster attack convergence by rapidly adapting to the deliberately induced misclassifications, accelerating the attack process. (iii) More effective exploitation of model weaknesses creating an inverse relationship between vulnerability and accuracy. Our F-CimBA can only infer the most confused class score (i.e., the prediction probability) as feedback because of its black-box nature. The aim is to guide F-CimBA to increase this misprediction probability with lower attack visibility by searching the random gradient perturbation.

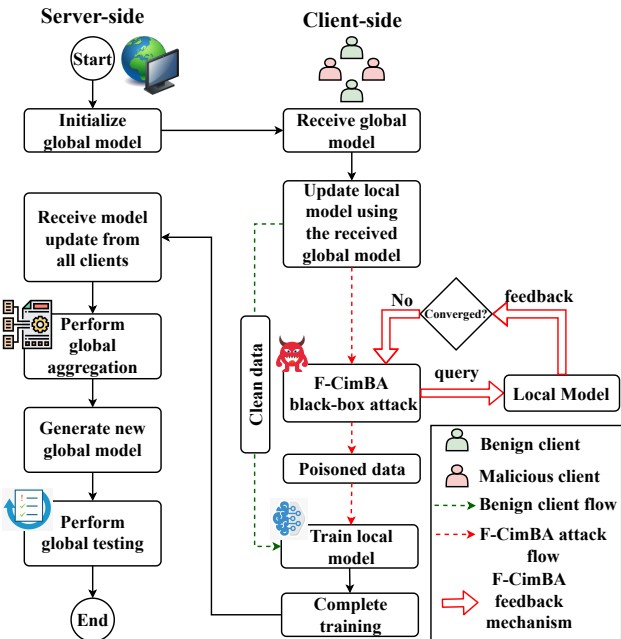

Figure 2: FL Flowchart with the proposed F-CimBA attack.

Let $\mathcal{X}$, $\mathcal{Y}$ be the image and label space, respectively, of the local client $\mathcal{C}_k$ such that $\mathcal{X} = \{\{\mathcal{X}_1\}_{i=1}^{B}, \ldots, \{\mathcal{X}_{B_n}\}_{i=1}^{B}\}$ and $\mathcal{Y} = \{\{\mathcal{Y}_1\}_{i=1}^{B}, \ldots, \{\mathcal{X}_{B_n}\}_{i=1}^{B}\}$, where each $\mathcal{X}_i$ is with a batch size of $B$ and $B_n$ is the total number of batches in $\mathcal{D}_k$. Since the gradient information is not available for the black-box model $f_\theta$, the output probabilities act as an effective proxy to direct the search for the perturbation that generates the adversarial image. In this untargeted attack scenario, the goal is to increase the probability of the most confused class. Given a clean image $\mathcal{X}_{l,i}$ and its corresponding label $\mathcal{Y}_{l,i}$, the F-CimBA adversarial client generates a random gradient perturbation $\mathcal{U}$, which is then scaled by a noise coefficient $\mu$, such that the prediction $f_\theta(\mathcal{X}_i + \mu \times \mathcal{U})$ does not match the label $\mathcal{Y}_i$, as depicted in Figure 3. The final adversarial image $\tilde{\mathcal{X}}_i$ is computed by adding the perturbation to the original image, i.e., $\tilde{\mathcal{X}}_i = \mathcal{X}_i + \mu \times \mathcal{U}$. The adversary uses the local black-box model on $\tilde{\mathcal{X}}_i$ to calculate the most confused class score (feedback to the F-CimBA), as feedback $= \max_{\tilde{\mathcal{Y}}_i \neq \mathcal{Y}_i}\{P(\tilde{\mathcal{Y}}_i|\tilde{\mathcal{X}}_i)\}$, where $\mathcal{Y}_i$ & $\tilde{\mathcal{Y}}_i$ are original & predicted classes, respectively. For the initial iteration, the random gradient perturbation ($\mathcal{U}$) is added in the positive direction. The gradient is added in the negative direction for further iterations and is changed to random perturbations in the subsequent iterations. The process of updating the random gradient perturbation $\mathcal{U}$ is repeated until the feedback converges and the algorithm generates the final adversarial image $\tilde{\mathcal{X}}_i$ and predicted label $\tilde{\mathcal{Y}}_i$.

In addition, F-CimBA converges on the $L2$ norm such that $||\tilde{\mathcal{X}} - \mathcal{X}|| \leq \mu$. The noise coefficient parameter $\mu$ controls the deviation of adversarial image w.r.t. original image without, making it perceivable to the human eye (helps avoid filtering images via human inspection). This step returns the final adversarial image to the client's dataset for further local model training via trusted execution environments (TEE) (Mondal et al., 2021; Chen et al., 2020). The entire F-CimBA algorithm is shown in Algorithm 1. In this work, we cannot directly compare the local model parameters due to the black-box attack settings. Therefore, we utilize the validation accuracy as a proxy metric to evaluate the visibility performance of the local models.

## 5 Convergence Analysis of F-CimBA

**Theorem 5.1** *Consider a federated learning system consisting of $n$ clients, among which $m$ clients are malicious and possess global model parameters $\theta_g$ (Definition 3.1). The global model output is given by $\mathcal{G}_{\theta_g}(\cdot)$, and the poisoned dataset generated by the proposed FFA method is denoted by $\tilde{\mathcal{D}}$. Let $\Delta\theta_{\tilde{G}}^t$ represent the aggregated adversarial model updates, and $\mu$ denote the noise coefficient. The proposed FFA method utilizes the most confused class score (feedback), defined as feedback $= \max_{\tilde{\mathcal{Y}} \neq \mathcal{Y}}\{P(\tilde{\mathcal{Y}}|\mathcal{X})\}$, where $\mathcal{Y}$ and $\tilde{\mathcal{Y}}$*

---

**Algorithm 1 Proposed F-CimBA attack method integrated FL process**

---

**Input:** Initialized global model parameters ($\theta_g^{\text{init}}$), local client training data ($\{\mathcal{D}_k\}_{k=1}^n$), noise coefficient ($\mu$)
**Output:** Global test accuracy after attack ($\mathcal{A}_\mathcal{G}^*$)

1: **function** Client execution($\mathcal{G}_{\theta_g}^t$)
2:     **for** $k = 1$ **to** $n$ **do**                                                              ▷ *Loop through n clients*
3:         **if Client type[k]** $== \mathcal{A}$ **then**
4:             Initialize the local model ($\tilde{f}_{\theta,k}^t$) parameters with shared global model parameters $\tilde{\theta}_k^t \leftarrow \theta_g^t$
5:             **Train local model $\tilde{f}_{\theta,k}^t$ on F-CimBA poisoned data to generate poisoned model parameters $\tilde{\theta}_k^t$**
6:             **for** $s = 1$ **to** samples $\in \mathcal{D}_k$ **do**
7:                 $\mathcal{U} =$ random perturbation noise
8:                 $\tilde{\mathcal{X}}_s = \mathcal{X}_s + \mu \times \mathcal{U}$                                      ▷ Initial adversarial image
9:                 $\tilde{\mathcal{Y}}_s = f_\theta(\tilde{\mathcal{X}}_s)$                                              ▷ Predicted label
10:                feedback$_{old}$ = $\max_{\tilde{\mathcal{Y}}_s \neq \mathcal{Y}_s}\{P(\tilde{\mathcal{Y}}_s|\tilde{\mathcal{X}}_s)\}$
11:                feedback$_{new}$ = feedback$_{old}$
12:                `iter` $\leftarrow$ Initialize max iterations
13:                **while** feedback$_{new} \leq$ feedback$_{old}$ and `iter` $!= 0$ **do**
14:                   $\tilde{\mathcal{X}}_s = \mathcal{X}_s + \mu \times \mathcal{U}$
15:                   $\tilde{\mathcal{Y}}_s = f_\theta(\tilde{\mathcal{X}}_s)$
16:                   feedback$_{new}$ = $\max_{\tilde{\mathcal{Y}}_s \neq \mathcal{Y}_s}\{P(\tilde{\mathcal{Y}}_s|\tilde{\mathcal{X}}_s)\}$
17:                   `iter` $-= 1$
18:                   $\mathcal{U} =$ update random perturbation noise
19:             $\tilde{\mathcal{D}}_k \leftarrow \{\tilde{\mathcal{X}}_s, \tilde{\mathcal{Y}}_s\}$
20:             $\tilde{\theta}_k^t \leftarrow \tilde{\theta}_k^t - \eta \nabla \mathcal{L}(\tilde{f}_{\theta,k}^t(\tilde{\mathcal{D}}_k))$
21:         **else**
22:             Initialize the local model ($f_{\theta,k}^t$) parameters with shared global model parameters $\theta_k^t \leftarrow \theta_g^t$
23:             Train local model on normal batch data for $E$ local epochs
24:             $\theta_k^t \leftarrow \theta_k^t - \eta \nabla \mathcal{L}(f_{\theta,k}^t(\mathcal{D}_k))$
25:             Scale up updates: $\theta_k^t \leftarrow \lambda_k \theta_k^t$
26:     **return** $\{\tilde{\theta}_k^t\}_{k=1}^m + \{\theta_k^t\}_{k=m+1}^n$
27: **function** Main( )
28:     $\theta_g^1 = \theta_g^{\text{init}}$                                                      ▷ *Initialize global model $\mathcal{G}_{\theta_g}^1$*
29:     **for** $t = 1$ to $T$ **do**                                        ▷ *Loop through T global epochs*
30:         $\{\theta_k^t\}_{k=1}^n \leftarrow$ CLIENT EXECUTION ($\mathcal{G}_{\theta_g}^t$)
31:         Update the global model: $\mathcal{G}_{\theta_g}^{t+1}$                      ▷ *Perform FedAvg as per Eq 1*
32:         $\mathcal{A}_\mathcal{G}^* \leftarrow$ `Test`($\mathcal{G}_{\theta_g}^{t+1}, \mathcal{D}_{test}$)             ▷ *Perform global model testing*
33:     **return** $\max(\mathcal{A}_\mathcal{G}^*)$

---

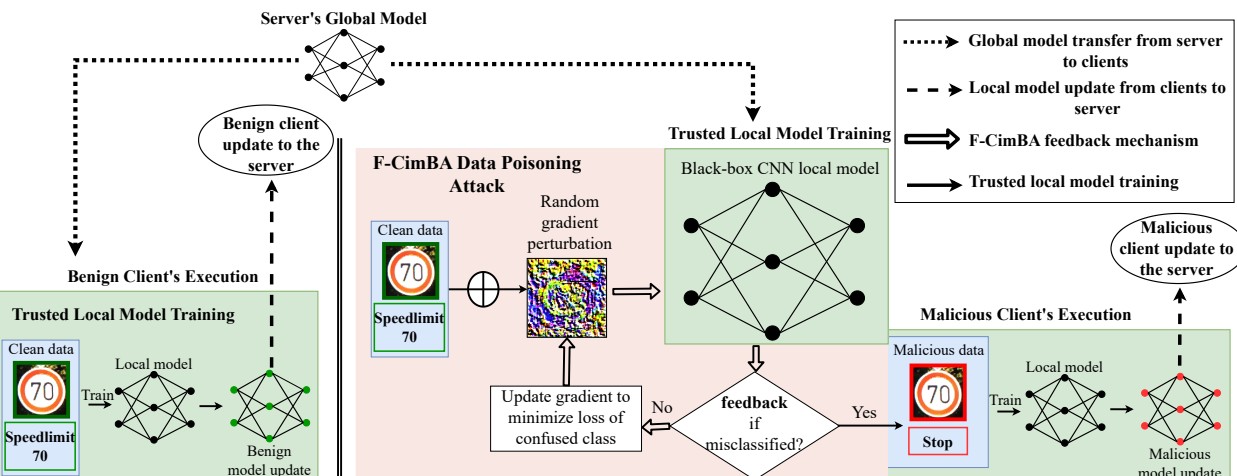

Figure 3: Proposed pipeline: The local training on clients happens in a trusted and isolated container where the attacker has no control over the local model and hence black-box. Benign clients train on clean data to generate regular updates, whereas malicious clients poison the training data using feedback-guided F-CimBA attack to generate malicious updates.

*are the original and predicted classes, respectively, to generate $\mu$-bounded adversarial data $(\tilde{\mathcal{X}})$ (Definition 4.1). Given the noise coefficient $(\mu)$, our F-CimBA attack converges to a $\delta$-bounded impact after $T$ rounds of communication as*

$$\frac{1}{T} \sum_{t=1}^{T} \mathcal{L}(\theta_g^t + \Delta\theta_{\tilde{\mathcal{C}}}^t, \mathcal{D}_{test}) - \mathcal{L}(\theta_g^t, \mathcal{D}_{test}) \leq \delta, \ s.t. \ \delta \geq \mu,$$

*where $\mathcal{L}(\theta_g^t, \mathcal{D}_{test})$ is the loss of the global model on the clean test dataset $\mathcal{D}_{test}$, and $\delta$ is bounded by the relation $\delta \geq \mu$.*

**Proof.** Let $\tilde{\mathcal{X}}_i$ be the adversarial image generated by the F-CimBA attack, and $\tilde{\mathcal{Y}}_i$ be the corresponding label. This process results in $\tilde{\mathcal{D}}_k$ poisoned training data. The $\mu$-noise coefficient makes sures that adversarial samples are $\mu$ bounded on to $L_2$ norm such that $\|\tilde{\mathcal{X}} - \mathcal{X}\| \leq \mu$ and the global model $\texttt{Val}_{\mathcal{G}^t}$ varies from that of the poisoned local model $\texttt{Val}_{\tilde{f}^t}$ for every $t \in T$. This constraint helps maintain the poisoned local model $\tilde{f}_{\theta,k}^t$ close to the global model $\tilde{\mathcal{G}}_\theta^t$, making it difficult for the server to detect the presence of poisoned data.

*Convergence criterion.* In classification tasks, the loss function is directly related to the attack objective, where a larger loss value increases the likelihood of misclassification (Gu et al., 2022). Consequently, in our FL setup, we establish the convergence criterion based on the global model's loss, which is directly connected to both the attack goal and convergence of our proposed F-CimBA method. The proposed FFA method generates $\mu$-bounded poisoned data, which constrains the impact of adversarial model updates, $\Delta\theta_{\tilde{\mathcal{C}}}^t$. After $T$ rounds of communication, the adversarial attack converges to a $\delta$-bounded impact, where $\delta$ depends on the perturbation magnitude $\mu$ such that a higher value of noise coefficient leads to a larger loss. Hence, we establish a relationship between $\delta$ and the $\mu$ parameter by considering the convergence criterion:

$$\frac{1}{T} \sum_{t=1}^{T} \mathcal{L}(\theta_g^t + \Delta\theta_{\tilde{\mathcal{C}}}^t, \mathcal{D}_{test}) - \mathcal{L}(\theta_g^t, \mathcal{D}_{test}) \leq \delta,$$

where $\mathcal{L}(\theta_g^t, \mathcal{D}_{test})$ is the loss of the global model on the clean test dataset. Taking into account the effects of $\mu$-bounded adversary, we can express the impact of poisoned data on the global model:

$$\Delta\mathcal{L} = \mathcal{L}(\theta_g^t + \Delta\theta_{\tilde{\mathcal{C}}}^t, \mathcal{D}_{test}) - \mathcal{L}(\theta_g^t, \mathcal{D}_{test}) \leq \mu.$$

This equation represents the maximum possible increase in the loss function due to poisoned data. Using the fact that $\Delta\mathcal{L} \leq \delta$, we establish a relationship between $\delta$ and the $\mu$ parameter as:

$$\delta \geq \mu.$$

This inequality indicates that $\delta$, the upper bound on the impact of poisoned data, should be greater than or equal to the noise coefficient $\mu$. This ensures that the attack converges while maintaining the desired $\mu$-separability constraint. Finally, the convergence is accomplished by leveraging model weaknesses through the most confused class score (feedback). This approach effectively maintains the perturbation magnitude within acceptable limits, ensuring the attack's convergence while efficiently generating adversarial examples. The update rule $\tilde{\mathcal{X}}_i = \mathcal{X}_i + \mu \times \mathcal{U}$ ensures that adversarial examples are generated by adding a scaled noise coefficient $\mu$ to the original input $\mathcal{X}_i$, providing a systematic method for creating poisoned samples.

## 6 Computational and Communication cost

The computational and communication costs of the F-CimBA-integrated FL system are discussed below.

**Lemma 6.1** *The expected time complexity of our proposed F-CimBA method is linear, specifically $\mathcal{O}(min(\tau, \gamma))$, where $\tau$ signifies the maximum iterations and $\gamma$ represents iterations required to find a solution ($\gamma <= \tau$).*

**Proof.** The proposed F-CimBA method generates an adversarial image through a novel query-based approach in a maximum of $\tau$ iterations. Hence, the notation $\mathcal{O}(min(\tau, \gamma))$ indicates that the complexity depends on the minimum of $\tau$ and $\gamma$ values. If $\gamma$ is smaller, a solution is found before completing all iterations ($\mathcal{O}(\gamma)$). If $\gamma$ equals $\tau$, the complexity is $\mathcal{O}(\tau)$, considering all iterations. Notably, F-CimBA time complexity is based on feedback, causing the variable time complexity $\mathcal{O}(min(\tau, \gamma))$.

The training process on the F-CimBA adversarial client takes approximately 10 seconds, with the duration varying based on the batch size in the training dataset. This procedure employs a single Nvidia Tesla M60 GPU with 8GB of RAM, using around 3.2GB, which highlights the low computational overhead.

*Communication cost-* As described in Algorithm 1, in each round, the malicious client communicates only model updates to the server, in line with the standard FL system process. Therefore, incorporating F-CimBA into the existing FL framework does not introduce significant additional communication overhead.

## 7 Experiments

In this section, we evaluate F-CimBA and compare its performance against state-of-the-art methods. Specifically, we address the following four major points to guide our evaluation:

1. Effectiveness of our F-CimBA method compared to state-of-the-art attacks.

2. F-CimBA's effectiveness under different degrees of non-IID data distribution.

3. Robustness of F-CimBA against inherent defense mechanisms, including Byzantine robust aggregation techniques and recent server-side defense methods.

4. Effectiveness of our F-CimBA method in minimizing attack visibility compared to other attack strategies.

### 7.1 Datasets and Metrics

We use the following four standard classification datasets and CNN models to demonstrate the diversity for our evaluation. Table 4 presents comprehensive experimental details of our evaluation.

- *GTSRB (Stallkamp et al., 2011).* is a well-known benchmark dataset for traffic sign classification. It consists of 43 traffic sign classes with 39209 samples. We build a custom 4-layer CNN architecture followed by two fully connected layers and treat this as a global model for this dataset. It takes an input image of size $150 \times 150$. Unless specified otherwise, we consider 40 total clients and select all client updates at every communication round.

- *KBTS (Mathias et al., 2013).* is another well-known benchmark dataset for traffic sign classification. It consists of 62 traffic sign classes with 6978 samples. We use the earlier custom 4-layer CNN architecture as a global model for this dataset. Similar to the above dataset, we consider 40 total clients and select all client updates at every communication round.

- *CIFAR10 (Krizhevsky et al., 2009).* This well-known benchmark dataset for classification contains 60,000 samples with ten different classes, namely, aeroplanes, cars, birds, cats, deer, dogs, frogs, horses, ships, and trucks. We use ResNet18 (He et al., 2015) architecture with input size as $224 \times 224$. We consider 100 total clients, out of which 40 client updates are randomly selected at every communication round.

- *EMNIST (Cohen et al., 2017).* This is another benchmark dataset of 671,585 samples of handwritten characters & digits with 62 classes, including upper and lowercase handwritten characters. We use LeNet (LeCun et al., 1998) architecture that takes an input of size $32 \times 32$. Further, we consider 10000 total clients, out of which 100 client updates are randomly selected at every communication round.

Table 4: Detailed experimental setup: including datasets, models, federated learning configuration, and attack parameters.

| Dataset(s) | GTSRB (Stallkamp et al., 2011) KBTS (Mathias et al., 2013) | CIFAR10 (Krizhevsky et al., 2009) | EMNIST (Cohen et al., 2017) |
|---|---|---|---|
| Model | 4 layered CNN
input (150 x 150 RGB images)
conv2d_64; kernel 5; stride 1
conv2d_128; kernel 3; stride 1
conv2d_256; kernel 1; stride 1
conv2d_256; kernel 1; stride 1
Fully connected layer 1
Fully connected layer 2
Softmax | ResNet18 (He et al., 2015) | LeNet (LeCun et al., 1998) |
| Total clients ($n$) | 40 | 100 | 10000 |
| Number of clients selected per round ($k$) | 40 | 40 | 100 |
| Total malicious clients ($m$) | 1, 8, 12, 16, 20, and 32 | 1, 10 | 1, 10 |
| Amount of non-IID ($\beta$) | 0.1, 0.5, 1 (default), 5, 10 | 0.1, 0.5, 1 (default), 5, 10 | 0.1, 0.5, 1 (default), 5, 10 |
| Batch size ($B$) | 64 | 64 | 64 |
| Global communication rounds ($T$) | 200 | 200 | 200 |
| Local epochs ($E$) | 5 | 5 | 5 |

We split all datasets into training ($\mathbb{D}_{Train}$, 80%) and test ($\mathbb{D}_{Test}$, 20%) sets. We further divide each of the clients' data into non-IID parts. Specifically, we split the training set using Dirichlet distribution (Minka, 2000) with parameter $\beta = 1$ (default). Later, we evaluated our method with five different $\beta = [0.1, 0.5, 1, 5, 10]$ values to justify its robustness. Increasing the Dirichlet distribution parameter, $\beta$ generates more dense IID data splits among the clients. Finally, the test set is used on the server side to compute the global model accuracy. We use two attacker settings, namely *single-client* and *multi-client*. We set one client as malicious and others as benign for a single-client attack scenario. To handle randomness, we use single-attack scenarios by different malicious clients, namely, $C_1$, $C_3$, $C_5$, $C_7$, and $C_9$. In multi-client attack settings, we consider five different percentages of adversaries (volume of attack), {20%, 30%, 40%, 50%, 80%} i.e., {8, 12, 16, 20, 32} randomly chosen clients as malicious for GTSRB and KBTS datasets evaluation. We set 10 randomly chosen clients as malicious for CIFAR10 and EMNIST datasets. We did not try to increase the number of malicious clients further, as that would increase the number of malicious updates and, thereby, attack visibility and dilute the motivation of this work. We empirically set 1000 maximum attack iterations and initialized $\mu = 0.5$ in our experiments. We conducted each experiment three times and presented the results and graphs as the average outcomes of these three simulations with an error bar.

**Attack baselines.** We have thoughtfully selected state-of-the-art attack methods based on their relevance and uptodatedness. These methods have been categorized into three distinct groups for our analysis: (i) recent FL data poisoning attacks, including data poisoning attack-static label flip (DPA-SLF) (Shejwalkar et al., 2022) and data poisoning attack-dynamic label flip (DPA-DLF) (Shejwalkar et al., 2022), (ii) recent FL model poisoning attacks, including little is enough (LIE) (Baruch et al., 2019) and Fang (Fang et al.,

2020), and (iii) robust ML data poisoning attack, such as simple black-box attack (SimBA) (Guo et al., 2019). This categorization enables us to thoroughly assess the current landscape of state-of-the-art methods in this field.

**Metrics.** We use the maximum classification global test accuracy ($GTA \in [0, 100]$) for all global epochs as an evaluation metric. It is given as is defined as $GTA = \frac{TP+TN}{|\mathbb{D}_{Test}|} \times 100$, where $TP$, $TN$ are true positives and true negatives given by the aggregated global model. The effectiveness of an attack is gauged by its capacity to substantially reduce the $GTA$, all while remaining undetected by the server.

### 7.2 F-CimBA Performance Comparison with Existing Attacks

**No Attack and No Defense Case:** We established a baseline accuracy for our FL configuration by evaluating it without any attacks or defense mechanisms for datasets such as GTSRB, KBTS, CIFAR10, and EMNIST. The baseline global test accuracy (GTA) ranged from 83.45% to 91.12%, as shown in Table 5 for non-IID data distribution with $\beta = 1$ as the default value. Notably, EMNIST exhibited slightly lower performance due to our FL setting, where data is distributed among 10,000 clients, and the server aggregates updates from only 100 randomly chosen clients. This results in reduced GTA due to low data availability at the clients and a high bias in the generated global model updates. Further insights into F-CimBA performance across varying degrees of non-IID are elaborated in subsequent sections.

Table 5: GTA (%) comparison for four datasets under non-IID distribution for no attack and no defense settings.

| non-IID ($\beta$) | GTSRB | KBTS | CIFAR10 | EMNIST |
|---|---|---|---|---|
| 0.1 | 87.26±0.83 | 87.62±0.24 | 81.36±0.20 | 79.65±0.27 |
| 0.5 | 88.12±0.38 | 89.14±0.54 | 85.34±0.38 | 82.67±0.63 |
| 1 (default) | 89.98±0.85 | 91.12±0.10 | 90.85±0.67 | 83.45±0.57 |
| 5 | 90.23±0.12 | 92.23±0.45 | 91.123±0.43 | 84.12±0.40 |
| 10 | 91.26±0.72 | 93.42±0.21 | 92.12±0.64 | 84.28±0.84 |

**F-CimBA Attack and No Defense Case:** We provide dataset-wise condensed highlights of our experimental results due to space constraints.

*(i) GTSRB dataset.* In the experiment with the GTSRB dataset, all 40 client updates were included for aggregation (Table 4). We observe that our F-CimBA method consistently outperformed other baselines with a substantial GTA drop, approximately 11% drop under a single malicious client and 83% drop with 32 malicious clients, compared to existing methods, as shown in Table 6 and Table 7. Notably, even with only one malicious client, F-CimBA outperforms LIE and Fang methods, which are the second-best methods. This highlights F-CimBA's efficacy in inducing poisoned data, causing a significant drop in global test accuracy. Additional insights into F-CimBA's impact on various malicious client IDs are in Figure 4a. F-CimBA's feedback-guided approach efficiently identifies gradient perturbations capable of confusing the global model with any of the non-targeted classes. Particularly in challenging scenarios, with an 80% attack percentage and 32 out of 40 clients exhibiting malicious behaviour, F-CimBA consistently achieves remarkable accuracy reductions of less than 10%, as illustrated in Figure 4c.

*(ii) KBTS dataset.* Similar to the GTSRB dataset, F-CimBA consistently outperforms baselines for the KBTS dataset. It achieves a significant GTA drop of approximately 10% under a single malicious client and an impressive 85% drop with 32 malicious clients, surpassing existing methods (Table 6 and Table 7). F-CimBA's effectiveness is slightly higher on GTSRB than KBTS in the single-client attack (Figure 4b), attributed to GTSRB's higher intra-class variability. However, in multi-client attacks, F-CimBA accuracy drops are more pronounced for KBTS than GTSRB (Figure 4d), likely due to KBTS's broader class space. This enables feedback-guided F-CimBA to collaboratively compromise the global model non-targeted, generating potent perturbations with less than 10% GTA reductions for multi-client attack settings.

*(iii) CIFAR10 dataset.* The server employs a random client selection process, aggregating from only 40 out of 100 available client updates in each round, as detailed in Table 4. This results in the inclusion of malicious

client updates in 81 out of 200 global epochs for the single-client attack setting and in all 200 epochs for the multi-client attack scenario. Building on the insights gained from our experiments with the GTSRB dataset, the F-CimBA method consistently exhibits superior performance compared to other baseline methods. In the single-client attack setting, it leads to an approximate GTA drop of 10%, while in the ten-client attack setting, the GTA drop is approximately 23%, outperforming other methods as presented in Table 6 and Table 8. However, it's worth noting that the lower rate of malicious client selection (81 out of 200 epochs) in the single-client attack setting, compared to the GTSRB dataset, contributes to a relatively lower performance of F-CimBA. Furthermore, in the multi-client attack scenario, where there are ten malicious clients on average, only 4 of these updates are selected on average for aggregation due to the random selection process. Despite this constraint, the F-CimBA attack still achieves a significant GTA drop of approximately 23%, surpassing the second-best method. This success can be attributed to the influence of the 30 benign client updates, which help mitigate the impact of the malicious updates introduced by F-CimBA.

*(iv) EMNIST dataset.* The server employs a random client selection process, aggregating from only 100 out of 10,000 available client updates in each round, as detailed in Table 4. This results in the inclusion of malicious client updates in 45 out of 200 global epochs for the single-client attack setting and 59 out of 200 for the ten-client attack scenario. The EMNIST dataset setting poses a particularly challenging scenario, characterized by a lower probability of selecting malicious clients in both the single and multi-client attack settings. However, our F-CimBA method consistently demonstrates superior performance, showcasing a Global Test Accuracy (GTA) drop of approximately 6% and 8% less than the no-attack baseline in the respective attack scenarios. Notably, in the EMNIST dataset, where data is distributed across 10,000 clients with a non-IID parameter ($\beta = 1$), each client receives only around 150 samples, presenting additional constraints. Despite these challenges, our F-CimBA method surpasses other approaches, emphasizing its high-impact nature.

In summary, our thorough assessment of F-CimBA, considering various $n$ and $k$ values across four diverse datasets, underscores its remarkable effectiveness in comparison to other baseline methods. Notably, even when faced with a low rate of malicious client updates, F-CimBA consistently outperforms all other baselines. This is particularly significant in scenarios involving a single malicious client, as F-CimBA exhibits a substantial impact, further highlighting its robustness as the adversary's knowledge increases and F-CimBA clients become more prevalent.

Table 6: Comparison of GTA (%) with existing attacks for four datasets under a single-client attack setting. **Bold** result indicates the best result (lower is better) for an attack setting. F-CimBA consistently showed better accuracy drops across all datasets.

| Attack | GTSRB | KBTS | CIFAR10 | EMNIST |
|---|---|---|---|---|
| DPA-DLF | 80.08±1.28 | 84.23±2.36 | 80.91±2.01 | 80.94±7.81 |
| DPA-SLF | 82.44±2.61 | 84.01±2.01 | 81.49±2.33 | 80.91±1.13 |
| LIE | 79.36±1.48 | 83.36±1.03 | 81.01±1.62 | 80.02±0.67 |
| Fang | 80.01±1.28 | 82.45±1.48 | 80.04±1.77 | 78.89±1.59 |
| SimBA | 80.98±1.32 | 83.55±1.57 | 80.94±2.28 | 81.95±1.09 |
| F-CimBA (Ours) | **78.43±1.42** | **81.60±1.63** | **79.62±1.13** | **77.23±1.42** |

Table 7: Comparison of GTA (%) with existing attacks for two datasets under multi-client attack setting. **Bold** result indicates the best result (lower is better) for an attack setting. F-CimBA consistently showed better accuracy drops across all datasets.

| Attack percentage (%)- m/n | GTSRB | | | | | | KBTS | | | | | |
|---|---|---|---|---|---|---|---|---|---|---|---|---|
| | DPA-DLF | DPA-SLF | LIE | Fang | SimBA | F-CimBA (ours) | DPA-DLF | DPA-SLF | LIE | Fang | SimBA | F-CimBA (ours) |
| 20- 8/40 | 75.93±2.04 | 76.29±0.31 | 73.62±2.24 | 72.38±1.94 | 75.80±0.74 | **70.81±0.26** | 82.98±1.34 | 84.25±0.79 | 61.36±0.15 | 55.48±0.76 | 62.41±0.24 | **51.75±1.30** |
| 30- 12/40 | 75.04±2.49 | 76.79±1.37 | 70.26±0.80 | 69.34±1.13 | 74.41±2.03 | **66.60±1.88** | 83.47±1.62 | 80.29±0.81 | 44.42±0.97 | 43.36±1.79 | 45.76±1.71 | **41.38±1.63** |
| 40- 16/40 | 73.70±2.17 | 76.62±0.30 | 68.42±1.33 | 67.26±1.07 | 70.49±1.89 | **63.35±0.39** | 42.37±1.99 | 48.36±0.12 | 38.42±0.88 | 39.38±2.34 | 39.76±1.84 | **35.43±0.49** |
| 50- 20/40 | 68.54±0.58 | 74.58±2.18 | 56.34±0.84 | 59.48±2.07 | 63.46±0.85 | **51.32±1.77** | 38.06±2.32 | 39.49±1.19 | 30.42±1.64 | 30.36±0.25 | 31.24±2.19 | **28.26±1.17** |
| 80- 32/40 | 37.46±0.92 | 55.30±0.95 | 31.45±0.69 | 27.16±1.82 | 40.43±0.54 | **6.38±0.55** | 28.24±0.42 | 25.36±2.41 | 12.64±1.60 | 11.12±1.24 | 15.32±0.17 | **6.85±0.40** |

Table 8: Comparison of GTA (%) with existing attacks for two datasets under multi-client (10A) attack setting. **Bold** result indicates the best result (lower is better) for an attack setting. F-CimBA consistently showed better accuracy drops across all datasets.

| Attack | CIFAR10 | EMNIST |
|---|---|---|
| DPA-DLF | 76.96±0.86 | 79.73±1.35 |
| DPA-SLF | 74.64±1.94 | 78.10±0.24 |
| LIE | 69.04±1.34 | 77.70±0.56 |
| Fang | 70.89±1.12 | 78.03±1.13 |
| SimBA | 72.63±1.21 | 79.90±0.83 |
| F-CimBA (ours) | **67.34±0.62** | **75.40±0.87** |

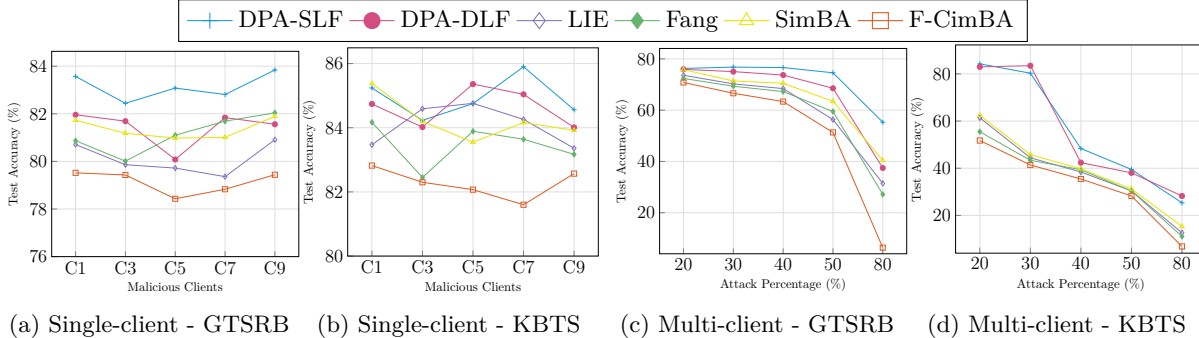

(a) Single-client - GTSRB    (b) Single-client - KBTS    (c) Multi-client - GTSRB    (d) Multi-client - KBTS

Figure 4: Effectiveness on F-CimBA against existing causative attacks: Comparison of GTA (%) for various baselines across two datasets and attack settings.

## 7.3 F-CimBA Performance Across Varying Degrees of non-IID

The influence of varying non-IID data distribution is a critical aspect that warrants further exploration. This examination allows us to better understand the interplay between the Dirichlet distribution (Minka, 2000) parameter $\beta$ and the resulting data distribution characteristics. The relationship between $\beta$ and the sample data partition is pivotal in comprehending the behavior of our experimental setup.

The Dirichlet distribution (Minka, 2000) serves as a key probabilistic model in Federated Learning (FL) to capture how data is distributed across multiple clients. This distribution is governed by a parameter $\beta$, which plays a crucial role in controlling the level of non-IIDness (non-independent and identically distributed nature) in the data partitioning process. The Dirichlet distribution generates data allocations for clients based on their specific characteristics, and the probability density function is given by: $p(x_1, x_2, \ldots, x_K|\beta) = \frac{1}{B(\beta)} \prod_{i=1}^{K} x_i^{\beta_i - 1}$, where $x_1, x_2, \ldots, x_K$ denote the data proportions assigned to each client, and $K$ represents the number of classes. The parameter vector $\beta = (\beta_1, \beta_2, \ldots, \beta_K)$ influences the distribution shape. In our context, we consider the case where all $\beta_i$ are identical, resulting in a symmetric Dirichlet distribution. The normalizing constant $B(\beta)$ is the multivariate Beta function, which ensures that the distribution is properly normalized across the simplex defined by the proportions. The Beta function is defined as: $B(\beta) = \frac{\prod_{i=1}^{K} \Gamma(\beta_i)}{\Gamma\left(\sum_{i=1}^{K} \beta_i\right)}$, where $\Gamma(\cdot)$ is the Gamma function. By adjusting the values of $\beta$, the distribution can either promote IID behavior, where data is evenly distributed across clients, or induce varying levels of non-IIDness. Specifically, larger values of $\beta$ tend to lead to a more even distribution of data across clients, while smaller values of $\beta$ create more skewed data allocations, amplifying heterogeneity among clients. This fine-tuning of $\beta$ is vital in FL systems, as it enables the model to reflect the real-world data heterogeneity present across clients, thereby influencing the robustness and generalization performance of the global model.

Our experimentation delves into the symbiotic relationship between the Dirichlet distribution parameter $\beta$ and FL attack dynamics. This interaction is pivotal for our study, as non-IID client datasets can significantly impact the global model's accuracy, even prior to the introduction of an attack. This pre-existing effect

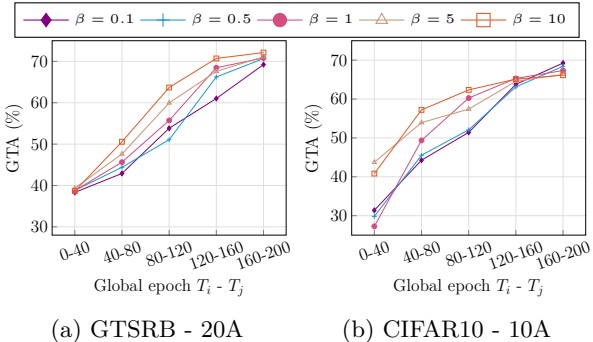

(a) GTSRB - 20A    (b) CIFAR10 - 10A

Figure 5: Analysis of varied degrees of Dirichlet distribution ($\beta$) non-IID on our F-CimBA attack impact for multi-client 20A and 10A across GTSRB , and CIFAR10 datasets, respectively.

arises due to biased and overfitted client models that emerge from non-IID local datasets. This phenomenon amplifies the overall attack impact, as demonstrated in Table 5.

However, it's important to recognize that the impact of non-IIDness is not solely governed by $\beta$. A confluence of factors, such as the total number of clients, the clients selected per round, and local and global training epochs, collectively influence the magnitude of the GTA under no attack scenarios. This amalgamation of parameters explains the substantial disparities in the low GTA values observed across datasets like CIFAR10, EMNIST, KBTS, and GTSRB. The intricate relationship between the number of clients, the selection per round, and the resultant GTA values become especially evident when analyzing scenarios with high non-IID data distribution, such as in the case of the $\beta = 0.1$ setting for the EMNIST dataset as shown in Table 5. It is worth noting that the EMNIST dataset exhibits a lower base performance than the other datasets, likely due to the distribution of data among the 10000 clients and the use of 100 randomly chosen clients for aggregation.

Furthermore, to evaluate the effectiveness of our proposed F-CimBA attack in highly non-IID settings, we conducted experiments on the GTSRB and CIFAR10 datasets using different $\beta$ values (0.1, 0.5, 1, 5, and 10) for multi-client attack 20A, 10A, respectively, as shown in Figure 5 at constant regular intervals of global epochs (for better visualization). The chosen $\beta$ values represented varying degrees of data distribution among clients, ranging from sparse and unbalanced to dense and balanced, as discussed above. Our findings demonstrate that the F-CimBA attack exhibits a high level of efficacy under non-IID data distributions. Specifically, for the GTSRB dataset with 40 clients, a $\beta$ value of 0.1 resulted in a lower GTA, while for the CIFAR10 dataset with 100 clients, a $\beta$ value of 0.1 yielded a lower GTA as well. Conversely, a $\beta$ value of 5 exhibited higher GTA for GTSRB and CIFAR10. This observation can be attributed to the interplay between the selection of malicious updates and the level of data distribution. Specifically, for GTSRB, where the malicious update is always chosen, the combination of $\beta = 1$ and 20% attack volume maximizes the F-CimBA impact. In contrast, for CIFAR10, where malicious updates are not always selected and with an increase in data due to $\beta = 5$, the F-CimBA poisoned data volume increases with an increase in attack volume. However, we note that further increasing $\beta$ leads to a drop in impact due to the influence of an increasing amount of benign data. Furthermore, the variance in the attack impact within a range of approximately $\pm 0.2$ can be attributed to the data poisoning nature of F-CimBA, where lower $\beta$ values result in fewer distributed and sampled data among clients. Our future investigation will focus on dynamically adjusting the attack budget in terms of attack volume for every $\beta$ distribution to enhance the attack impact.

In summary, our in-depth analysis of the non-IID data distribution's impact on FL attacks provides vital insights into the complex dynamics governing FL system performance. The careful calibration of $\beta$ and its repercussions on data distribution elucidate the underlying factors that can lead to substantial variations in model accuracy and attack effectiveness. This exploration enriches our understanding of FL's behavior under varying conditions and underscores the importance of accounting for non-IIDness in practical scenarios.

### 7.4 F-CimBA's Resilience against Defenses

We have carefully chosen the following state-of-the-art defense methods due to their relevance and recent advancements in the field. Subsequently, we have organized them into two distinct categories for our analysis. The first category encompasses Byzantine robust aggregation techniques, including Krum (Blanchard et al., 2017), Trimmed mean (Yin et al., 2018), and Median (Yin et al., 2018). The second category comprises very recent defense methods, such as FLTrust (Cao et al., 2021), and even those introduced in 2023, like LoMar (Li et al., 2023) and FL-Defender (Jebreel & Domingo-Ferrer, 2023). We have organized these techniques in this way to enable a thorough assessment of the robustness of our F-CimBA against the latest state-of-the-art defense methods in this domain. Below, we provide further details about these defenses.

1. *Krum (Blanchard et al., 2017):* Selects a representative local model update by evaluating the majority of client models. In our experiments, we set $x = 10$.

2. *Trimmed mean (Yin et al., 2018):* Performs aggregation of input updates by discarding the $x$ largest and smallest values in each dimension. We use $x = 2$ for our setup.

3. *Median (Yin et al., 2018):* Aggregates the updates by computing the median of each dimension.

4. *FLTrust (Cao et al., 2021):* Utilizes an auxiliary model trained on a root dataset to compute client trust scores, which are based on the similarity of each client's weight updates to the global model. The server updates are then weighted according to these trust scores.

5. *LoMar (Li et al., 2023):* Employs kernel density estimation to evaluate model updates and identifies an optimal threshold for distinguishing between malicious and legitimate updates.

6. *FL-Defender (Jebreel & Domingo-Ferrer, 2023):* Analyzes the behavior of neurons, proposes robust discriminative features, compresses similarity vectors and adjusts worker updates' weights prior to aggregation.

For a comprehensive evaluation, we conducted comparative analyses of our proposed F-CimBA attack method against the best-performing baseline attacks from various categories, including DPA-DLF, SimBA, and LIE. Our assessment primarily focused on evaluating the resilience of the F-CimBA attack when exposed to defense mechanisms. Specifically, we considered scenarios involving single-client attacks on the GTSRB and KBTS datasets, as well as multi-client attacks (10A) on the CIFAR10 dataset. In the single-client attack settings, the server selected all 40 clients for aggregation in the case of the GTSRB and KBTS datasets. In the multi-client attack scenario, the server randomly selected 40 out of 100 clients for aggregation, with ten of them being malicious clients. These configurations provided an optimal context to gauge the robustness of our F-CimBA attacks against contemporary defense mechanisms. Our experiments maintained default settings for other parameters, including the use of a $\beta = 1$ non-IID parameter and other parameters. The comparative results showcasing F-CimBA's performance against existing defenses are detailed in Table 9, Table 10, and Table 11.

We observed that our F-CimBA attack exhibited notable performance in various settings. Under the single-client attack scenario without any defense, we achieved an accuracy of 78.43% for the GTSRB dataset and 81.60% for the KBTS dataset. In the multi-client attack (10A) setting on the CIFAR10 dataset, we obtained an accuracy of 67.34% in the absence of any defense mechanism. When defense mechanisms were introduced, the accuracy remained competitive, reaching 75.12% for the GTSRB dataset under the LoMar defense and improving to 83.34% for the KBTS dataset under the FLDefender defense. Notably, compared to alternative methods, F-CimBA consistently performed better in terms of maintaining a lower GTA under attack conditions. For instance, the SimBA method achieved an accuracy of 81.16% (second-best) under attack. These findings align with our visibility analysis, discussed in subsequent sections, reinforcing the assertion that F-CimBA offers lower attack visibility compared to its counterparts. These trends persisted in the case of multi-client F-CimBA attacks (10A) even when defenses were in place. Under the LoMar defense, GTA improved slightly to 75.91%, whereas the second-best attack method achieved a GTA of 77.42%. This consistent observation underscores the robustness of the F-CimBA attack, particularly as a feedback-guided

Table 9: Comparison of GTA (%) of F-CimBA with other attack methods under state-of-the-art defense techniques for the GTSRB dataset. We use the 1A attack setting for this experiment. **Bold** result indicates the best results.

| Defense | DPA-DLF | SimBA | LIE | F-CimBA (ours) |
|---|---|---|---|---|
| Krum | 79.45±1.27 | 79.91±0.98 | 80.24±0.12 | **73.95±1.41** |
| TM | 76.42±1.38 | 75.87±1.15 | 78.36±0.14 | **69.98±1.49** |
| Median | 78.92±0.68 | 76.85±1.91 | 79.42±0.93 | **72.71±0.58** |
| FLTrust | 8.26±1.75 | 8.26±1.75 | 8.26±1.75 | **8.19±0.84** |
| LoMar | 80.45±2.30 | 81.16±1.87 | 81.13±1.59 | **75.12±0.39** |
| FLDefender | 82.34±0.57 | 80.23±1.29 | 82.16±0.81 | **74.36±1.07** |

Table 10: Comparison of GTA (%) of F-CimBA with other attack methods under state-of-the-art defense techniques for the KBTS dataset. We use the 1A attack setting for this experiment. **Bold** result indicates the best results.

| Defense | DPA-DLF | SimBA | LIE | F-CimBA (ours) |
|---|---|---|---|---|
| Krum | 83.32±1.45 | 85.19±0.83 | 83.49±1.76 | **82.98±0.81** |
| TM | 80.19±1.05 | 83.98±0.69 | 82.26±1.46 | **73.46±1.19** |
| Median | 79.94±0.18 | 85.19±1.20 | 82.45±0.33 | **74.98±1.77** |
| FLTrust | 10.24±2.12 | 10.24±2.12 | 10.24±2.12 | **9.65±1.95** |
| LoMar | 84.26±1.24 | 84.98±0.15 | 84.48±0.47 | **82.16±1.72** |
| FLDefender | 84.49±0.57 | 85.46±0.72 | 85.37±1.46 | **83.34±0.41** |

approach capable of evading server-side defense mechanisms. Notably, the FLTrust method, reliant on an auxiliary model trained using a root dataset provided by the server, exhibited poor performance across all attacks, indicating that its limited representation of client data in the root dataset adversely affected its effectiveness.

Overall, the F-CimBA attack demonstrated low attack visibility and proved to be a resilient attack method even when defenses were deployed. It exhibited only marginal improvements in GTA, approximately 2% and 8% for single and multi-client F-CimBA attacks, respectively, whereas other methods displayed vulnerability to the defense mechanisms.

## 7.5 F-CimBA Method Attack Visibility Analysis

To assess the attack visibility of our F-CimBA attack, we conducted an experiment using the CIFAR10 dataset in a single-client attack scenario, employing default parameter settings (as detailed in Table 4). This dataset was selected due to its suitability for a comparative analysis of F-CimBA's visibility in relation to other attack methods, given that the server selects 40 malicious clients from a pool of 100. To ensure a fair and meaningful comparison, we selected the best-performing baseline attack from each category, namely DPA-DLF, SimBA, and LIE. In this visibility analysis, the malicious client calculated the validation accuracy of the updated local model after training and subsequently compared it to the accuracy of the current global model received from the server. The measure of attack visibility was quantified by calculating the absolute difference between these two accuracies, with a lower difference indicating reduced attack visibility. This experimental setup allowed us to comprehensively evaluate how effectively F-CimBA operates in terms of its visibility compared to state-of-the-art baseline attack methods. Figure 6 illustrates a comparative analysis of attack visibility, highlighting the performance of our F-CimBA attack in relation to other baseline methods. Our findings reveal that the F-CimBA attack exhibits significantly lower visibility compared to alternative methods, aligning with our expectations. This reduced visibility is attributed to F-CimBA's strategy of minimizing the loss associated with the most confused class through feedback-guided gradient perturbations.

Specifically, the DPA-DLF attack, characterized by label flipping, displays relatively second lower visibility since it doesn't directly manipulate the data. Our analysis reveals that model poisoning techniques like the LIE attack demonstrate notably high visibility, evidenced by significant disparities in validation accuracy. While our F-CimBA approach shows marginally elevated visibility during early rounds when local models

Table 11: Comparison of GTA (%) of F-CimBA with other attack methods under state-of-the-art defense techniques for the CIFAR10 dataset. We use the 10A attack setting for this experiment. **Bold** result indicates the best results.

| Defense | DPA-DLF | SimBA | LIE | F-CimBA (ours) |
|---------|---------|-------|-----|----------------|
| Krum | 77.91±2.41 | 79.62±1.02 | 80.64±1.15 | **75.45±0.21** |
| TM | 77.70±1.32 | 80.12±0.16 | 81.18±1.06 | **73.97±1.12** |
| Median | 77.51±1.27 | 81.23±0.23 | 80.50±2.12 | **74.23±1.13** |
| FLTrust | 9.49±1.53 | 9.49±1.53 | 9.49±1.53 | **8.23±1.21** |
| LoMar | 78.36±1.49 | 77.42±2.31 | 80.62±1.05 | **75.91±1.12** |
| FLDefender | 78.51±2.30 | 77.28±1.41 | 80.34±2.70 | **74.42±1.30** |

are learning data representations, this effect diminishes as the local model reaches stability. Specifically, the validation accuracy differential drops to less than 0.05 once stabilized. This stands in stark contrast to baseline approaches such as DPA-DLF, SimBA, and LIE, which exhibit inconsistent visibility patterns that generally increase over time. These findings demonstrate F-CimBA's superior ability to maintain low attack visibility compared to existing attack methodologies.

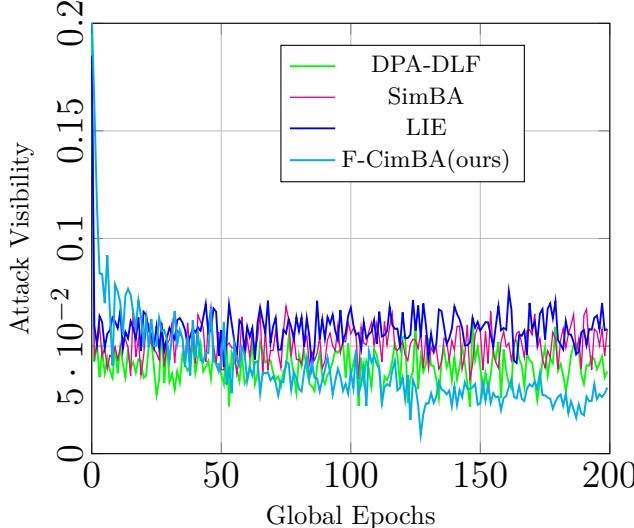

Figure 6: Comparison of the visibility of the F-CimBA attack with various baseline methods across different global epochs for the CIFAR-10 dataset under 1A settings.

To sum up, our F-CimBA approach has showcased superior performance in terms of lower attack visibility when compared to established baseline methods. This reduced visibility is attributed to F-CimBA's utilization of feedback-driven gradient perturbation techniques, which focus on minimizing the loss associated with the most confused class. These observations are substantiated by the results presented in the preceding section, where F-CimBA consistently outperformed other attacks, even when confronted with server-side defenses. Essentially, this implies that the perturbations introduced by F-CimBA are less conspicuous and pose a more challenging detection problem compared to the perturbations generated by other attack methods.

## 7.6  F-CimBA vs. Hardware Performance Metrics

This section analyzes the impact of the F-CimBA attack on system resources. Previous research has focused on traditional attack efficacy metrics such as global test accuracy, but this study is the first to examine hardware metrics. We analyze six metrics (execution time, bandwidth, CPU/GPU utilization and memory usage, and GPU temperature) under no attack, single-client attack, and multi-client (50%) attack across global epochs. This section investigates attack progression through the lens of system resource consumption.

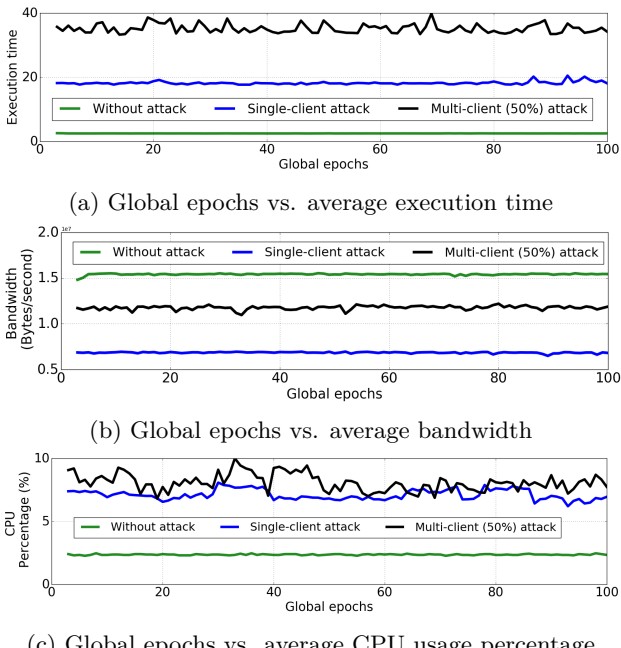

(a) Global epochs vs. average execution time

(b) Global epochs vs. average bandwidth

(c) Global epochs vs. average CPU usage percentage

Figure 7: Hardware performance graphs under (a) no attack, (b) single-client, and (c) multi-client attack using the proposed F-CimBA

Our motivating factor is understanding the bottleneck issues due to the F-CimBA attack. Prior works have emphasized the conventional attack efficacy metrics, such as the drop in the global test accuracy. To the best of our knowledge, this additional viewpoint of hardware metrics has never been presented before. We consider seven metrics, namely, *execution time, bandwidth (throughput of the uplink channel), CPU utilization, RAM utilization, GPU memory usage, GPU utilization, and GPU temperature*, in our evaluation. All the metrics are analyzed thoroughly under three settings: without attack, single-client attack, and multi-client (50%) attack throughout global epochs.

Table 12 and Figure 7 present the above-mentioned results. We observe that the attack drastically impacts the low-end resource-constrained client devices that can eventually become stragglers. Although the attack originated from a single/group of malicious clients in the minority, it eventually starts influencing the learning operations of all the benign clients. This perpetuates the attack even further as long as the minor perturbations remain undetected. We summarize our observations as ($i$) execution time increases up to $30\times$ when we compare multi-client attack w.r.t. no attack in Figure 7a. This drastically reduces the communication bandwidth in Figure 7b as it is calculated as the volume of information that can be sent over the communication channel in a measured amount of time. Also, it drastically increases the load on CPU utilization compared to without attack scenario, as shown in Figure 7c. Further, as expected, we observe the system hardware impact is much worse in multi-client w.r.t. single-client attacks. ($ii$) The overall GPU memory and utilization by the client and the central server remain consistent (Table 12 columns 2 & 3), irrespective of single or multi-client attacks. This is in stark contrast to variations in the CPU metrics, which show that the GPU is resilient under attack. ($iii$) The temperature of GPU hardware (Table 12 column 4) drastically rises under attack, leading to overheating and faster disruption. Overall, FL researchers working on adversarial attack detection can use these hardware performance trends to build future resilient FL systems.

Table 12: GPU memory (GM), GPU utilization (GU), GPU temperature (GT) under no attack, single-client, and multi-client attack using the proposed F-CimBA.

| Attack setting | GM (MB) | GU (%) | GT ($^\circ C$) |
|---|---|---|---|
| No attack | 5261 | 97 | 40 |
| Single-client attack | 5827 | 32 | 44 |
| Multi-client (50%) attack | 5827 | 32 | 46 |

## 8 Ablation Study

In this section, we evaluate the effectiveness of the F-CimBA attack against only SimBA (second best-performing attack) for brevity under client scaling, homogeneous & heterogeneous FL, attack progression w.r.t. global epochs, and convergence rate.

### 8.1 Client Scaling and Homogeneous & Heterogeneous FL Settings

To evaluate the impact of client scaling on the F-CimBA attack, we progressively increase the number of clients $n = 3, 5, 10, 15, 25, 40$ to present a controlled view of the experiment space for GTSRB and KBTS datasets (for brevity). Next, we consider another FL data shard setting called *homogeneous*, where $|\mathbb{D}_k| = l_k$ (dataset equally divided among clients) and show a comparative evaluation with heterogeneous FL settings, where $|\mathbb{D}_1| = l_1$, $|\mathbb{D}_2| = l_2$, ..., $|\mathbb{D}_n| = l_n$, where we assign data randomly such that $1000 \leq l_k \leq 1500$. Finally, we perform single-client and multi-client attacks with attack percentages as $\{30, 50\}\%$ for brevity under these different clients and FL settings. Along with the GTA (%), we also consider the number of global epochs for convergence and average attack success rate as metrics to compare the overall effectiveness.

**No attack and no defense case:** It is important to evaluate the "no attack and no defense" case in an experimental setting because it provides a baseline against which the effectiveness of different attacks and defenses can be measured. For example, if a defense is shown to improve the performance of the FL system under attack, it is important to know how much the attack degraded the FL system's performance in the first place. Without a baseline, it would be difficult to accurately assess the effectiveness of the defense. Evaluating the "no attack and no defense" case also help to ensure that any observed changes in performance are due to the attacks or defenses being tested and not other factors. This can increase the validity and reliability of the experimental results. Table 13 presents the minimum (40 clients) and maximum (3 clients) accuracies achieved under no attack and no defense ($BA$) results for both datasets. As we increase the number of clients, the data size of each client reduces, resulting in an accuracy drop. Further, homogeneous FL with the same amount of data has shown slightly better performance than heterogeneous FL settings for both datasets.

Table 13: Global test accuracy (%) under no attack and no defense techniques for both datasets' homogeneous and heterogeneous FL settings.

| | FL-Hom | | FL-Het | |
|---|---|---|---|---|
| | Max ($3C$) | Min ($40C$) | Max ($3C$) | Min ($40C$) |
| **GTSRB** | 94.8 | 90.23 | 95.2 | 89.8 |
| **KBTS** | 98.3 | 93.46 | 97.1 | 91.12 |

**Attack cases:** Figure 8 shows single-client attack accuracy, convergence results under different clients, and homogeneous & heterogeneous FL for both datasets. Interestingly, as we increase the number of clients, the number of benign clients increases, increasing overall accuracy. Further, F-CimBA-Het has shown better accuracy than SimBA attack configurations, as shown in Figure 8a and Figure 8b.

Furthermore, homogeneous FL showed a better convergence rate (less number of global epochs to converge) for both attacks and both datasets, as shown in Figure 8c, Figure 8d, and Figure 9. This is because each client has an equal data size, making it easier for the attack to generate an equal amount of poisoned data as benign data. This leads to similar malicious contributions and dominates the benign updates, leading to

early convergence. Further, F-CimBA configurations can converge faster than SimBA for single and multi-client attacks. Additionally, Figure 10 presents results for multi-client attacks with attack percentages of 30% and 50% for both datasets. We observe similar trends of single-client attack with reduced accuracies, and F-CimBA configurations outperformed other methods.

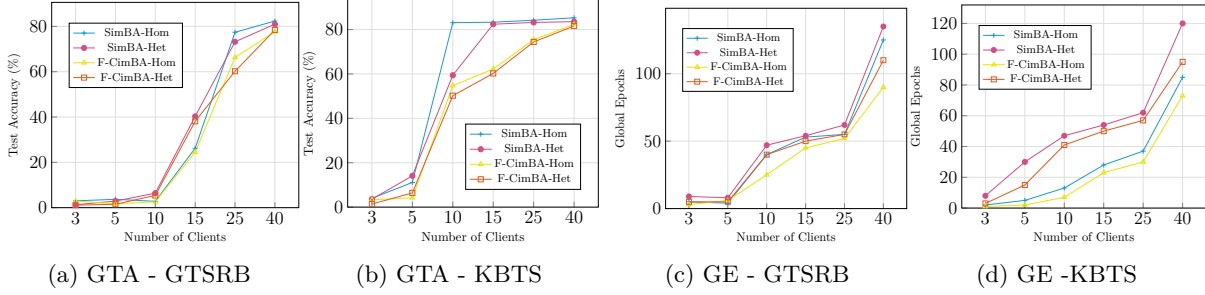

(a) GTA - GTSRB  (b) GTA - KBTS  (c) GE - GTSRB  (d) GE -KBTS

Figure 8: Comparison of global test accuracy (GTA) and global epochs (GE) of F-CimBA against SimBA across two different FL settings and single-client attack.

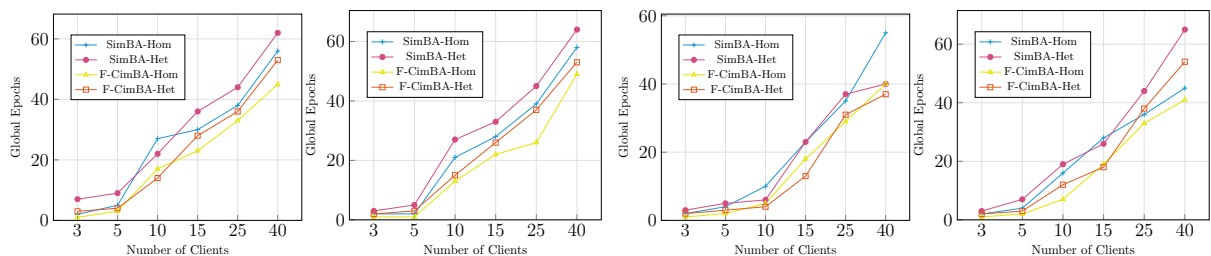

(a) 30% Malicious - GTSRB (b) 50% Malicious - GTSRB  (c) 30% Malicious - KBTS  (d) 50% Malicious - KBTS

Figure 9: Comparison of global convergence test epochs of F-CimBA against SimBA (second best-performing attack) across two different homogeneous & heterogeneous FL settings, and 30% & 50% multi-client attack.

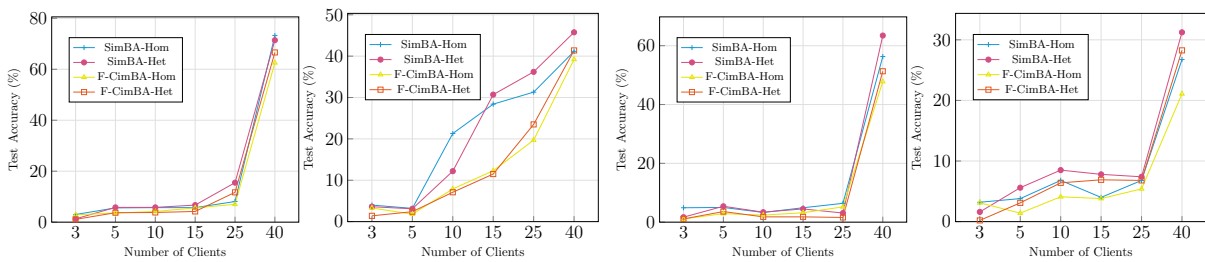

(a) 30% Malicious - GTSRB (b) 50% Malicious - GTSRB  (c) 30% Malicious - KBTS  (d) 50% Malicious - KBTS

Figure 10: Comparison of global test accuracy of F-CimBA against SimBA (second best-performing attack) across two different homogeneous & heterogeneous FL settings, and 30% & 50% multi-client attack.

*Average attack success rate:* Table 14 and Table 15 present results of the average attack success rate given by the ratio of the number of corrupted training samples per total samples for different F-CimBA configurations against SimBA configurations. We observe that for 50% of all cases, our F-CimBA has shown a 100% average attack success rate from low to larger client sizes.

*In summary, the feedback-guided mechanism in F-CimBA helps in early attack convergence, high success rate, and dropping the accuracy under single and multi-client attack settings for both datasets.*

Table 14: Average attack success rate (%) on GTSRB.

| Clients | SimBA-Hom | F-CimBA-Hom | SimBA-Het | F-CimBA-Het |
|---------|-----------|-------------|-----------|-------------|
| 3 | 99.38 | 100 | 96.92 | 98.46 |
| 5 | 100 | 100 | 95.23 | 98.41 |
| 10 | 91.83 | 95.91 | 97.29 | 97.29 |
| 15 | 96.87 | 100 | 93.54 | 96.77 |
| 25 | 89.47 | 94.73 | 94.11 | 100 |
| 40 | 90.23 | 95.68 | 92.88 | 98.97 |

Table 15: Average attack success rate (%) on KBTS.

| Clients | SimBA-Hom | F-CimBA-Hom | SimBA-Het | F-CimBA-Het |
|---------|-----------|-------------|-----------|-------------|
| 3 | 96.55 | 100 | 89.47 | 94.73 |
| 5 | 88.23 | 94.11 | 87.5 | 93.75 |
| 10 | 87.5 | 100 | 88.5 | 90.23 |
| 15 | 87.63 | 91.45 | 90 | 94.5 |
| 25 | 86.66 | 97.45 | 86.6 | 90.1 |
| 40 | 90.13 | 94.56 | 91.33 | 97.645 |

## 8.2 Byzantine Robust Analysis of F-CimBA

FL's Byzantine aggregation rules protect against malicious clients in Byzantine failures. These rules aggregate model updates in a way that resists the influence of malicious updates, improving the reliability and robustness of the FL system and protecting against attacks or other failures that could compromise the model's accuracy. We present more results of F-CimBA against Byzantine robust aggregation rules as an extension to Section 7.4.

**No attack with defense case:** We perform experiments on our FL setup with no attack with different byzantine robust aggregation rules at the servers as a base case to compare the F-CimBA attack accuracy drop and demonstrate the effectiveness. Table 16 presents the no attack with defense results for both datasets. We observe that the Krum method results are better than those of other methods under no attack.

Table 16: Global test accuracy (%) of our FL setting under no attack and with Byzantine robust defense techniques.

|  | GTSRB | KBTS |
|--------|-------|-------|
| **Krum** | 87.35 | 89.72 |
| **TM** | 85.43 | 87.34 |
| **Median** | 86.22 | 88.23 |

**Attack cases:** Figure 11 and Table 17 present the results (*AD*) of three different aggregation rules under single and multi-client attack settings for both datasets. *Single-client attack:* We observe that our proposed F-CimBA constantly showed robustness to byzantine aggregation rules compared to SimBA attack with an accuracy drop of around 13.4%, 17.37%, and 14.64% corresponding to Krum, TM, and median aggregation rules, respectively, for GTSRB dataset. For the KBTS dataset, the reported drop in accuracies are 6.74%, 16.26%, and 14.74% for Krum, TM, and Median aggregation rules, respectively, as shown in Figure 11a, Figure 11b, and Table 17. It is interesting to note that among the different Byzantine aggregation rules, Krum performed better than TM and median. This is because Krum selects one local model update representative of most client models by computing the pairwise distances between individual models (Blanchard et al., 2017). However, when the data across the workers are highly non-i.i.d. and poisonous, there is no 'representative' client model. Hence, it results in a performance drop under attack. On the other hand, trimming model updates after sorting client updates, including median, results in considering malicious updates for aggregation and hence the poor results.

*Multi-client attack:* Figure 11c, Figure 11d, and Table 17 present results on multi-client attack results with Byzantine robust aggregation rules. We observe a similar trend as single-client attack with a significant

drop in accuracy (rating from 20.38% - 84.63% for GTSRB and 30.3% - 82.87% for KBTS dataset) with the increase in attack percentage (from 20% - 80%). Further, TM has consistently shown less Byzantine robustness to our F-CimBA attack than other aggregation rules for GTSRB. In addition, on KBTS data, both TM and median have shown less performance than the Krum rule.

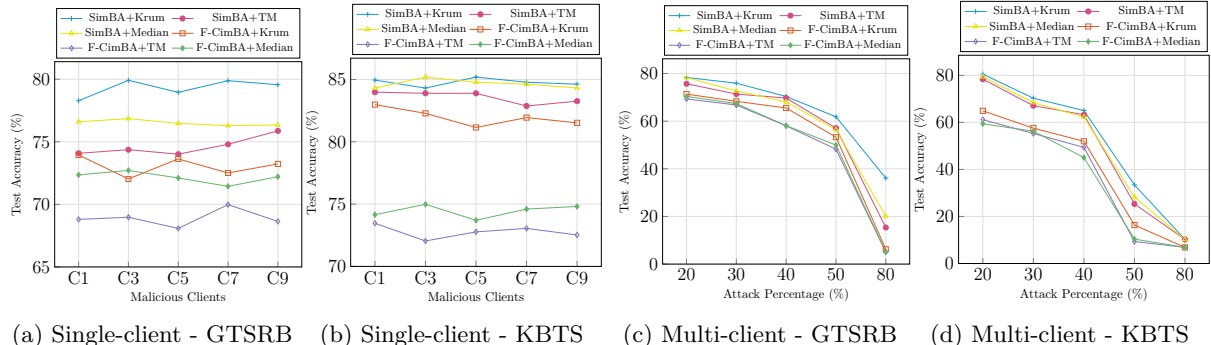

(a) Single-client - GTSRB  (b) Single-client - KBTS  (c) Multi-client - GTSRB  (d) Multi-client - KBTS

Figure 11: Effectiveness on F-CimBA against existing Byzantine robust aggregation techniques: Comparison of global test accuracy F-CimBA against SimBA (second best performing attack) across two different datasets and attack settings.

Table 17: Comparison of F-CimBA accuracy (%) with SimBA (second best performing attack) under Byzantine robust aggregation techniques for two datasets and attack settings. **Bold** result indicates the best results.

| | | GTSRB | | | | | | KBTS | | | | | |
|---|---|---|---|---|---|---|---|---|---|---|---|---|---|
| | Client Ids | SimBA+ Krum | SimBA+ TM | SimBA+ Median | F-CimBA+ Krum | F-CimBA+ TM | F-CimBA+ Median | SimBA+ Krum | SimBA+ TM | SimBA+ Median | F-CimBA+ Krum | F-CimBA+ TM | F-CimBA+ Median |
| **Single-client Attack** | C1 | 78.28 | 74.08 | 76.6 | 73.95 | **68.8** | 72.36 | 84.95 | 83.98 | 84.31 | 82.98 | **73.46** | 74.15 |
| | C3 | 79.91 | 74.37 | 76.85 | 72.03 | **68.97** | 72.71 | 84.31 | 83.89 | 85.19 | 82.28 | **72.04** | 74.98 |
| | C5 | 78.95 | 74.01 | 76.47 | 73.63 | **68.08** | 72.11 | 85.19 | 83.89 | 84.78 | 81.15 | **72.77** | 73.7 |
| | C7 | 79.88 | 74.8 | 76.29 | 72.51 | **69.98** | 71.44 | 84.78 | 82.87 | 84.62 | 81.94 | **73.05** | 74.6 |
| | C9 | 79.56 | 75.87 | 76.34 | 73.23 | **68.64** | 72.21 | 84.62 | 83.26 | 84.32 | 81.52 | **72.52** | 74.81 |
| | Attak Percentage (%) | SimBA+ Krum | SimBA+ TM | SimBA+ Median | F-CimBA+ Krum | F-CimBA+ TM | F-CimBA+ Median | SimBA+ Krum | SimBA+ TM | SimBA+ Median | F-CimBA+ Krum | F-CimBA+ TM | F-CimBA+ Median |
| **Multi-client Attack** | 20 | 78.35 | 75.71 | 78.28 | 71.36 | **69.34** | 70.38 | 80.5 | 78.36 | 79.36 | 64.92 | 61.18 | **59.42** |
| | 30 | 75.94 | 71.38 | 72.71 | 68.32 | **66.8** | 67.38 | 70.29 | 67.12 | 68.42 | 57.54 | **55.29** | 56.24 |
| | 40 | 70.36 | 69.71 | 68.07 | 65.54 | **58.05** | 58.11 | 65.04 | 63.25 | 62.31 | 51.97 | 49.38 | **45.04** |
| | 50 | 61.83 | 57.09 | 56.47 | 53.28 | **48.25** | 49.97 | 33.42 | 25.33 | 28.26 | 16.35 | **9.35** | 10.38 |
| | 80 | 36.11 | 15.36 | 20.04 | 6.23 | **5.09** | **5.09** | 10.23 | 10.23 | 10.23 | 6.85 | **6.85** | **6.85** |

In summary, F-CimBA is robust against Byzantine aggregation rules at the server for both attack settings and datasets. It has shown better robustness compared to the SimBA attack. With an ability to select a representative of most client models by computing pairwise distances, the Krum aggregation rule effectively shows better robustness than TM and median.

## 8.3  Time-series Analysis of F-CimBA Attack

Figure 12 displays results for test accuracy in regular and attack cycles for different clients for *heterogeneous* FL settings for the GTSRB dataset. We demonstrate only single-client attack results of our F-CimBA (Figure 12b) against SimBA (Figure 12a) for brevity. An expected, rising trend of test accuracy is observed for FL when trained on normal data. The model trained on normal data with three clients resulted in an accuracy of 95.2% for the GTSRB dataset under normal heterogeneous FL settings. The *attack phase* begins after completing the *normal phase* at 30 epochs. The normal and attack phases are juxtaposed to represent their contrasting behavior. We can observe a *significant distortion in the model test accuracy throughout the attack phase for the entire experiment design space* in all our result figures. Our proposed F-CimBA attack approach demonstrates consistent results in better convergence for heterogeneous clients (3, 5, 10) and overall average accuracy drop, causing maximum damage with a better attack local minimum.

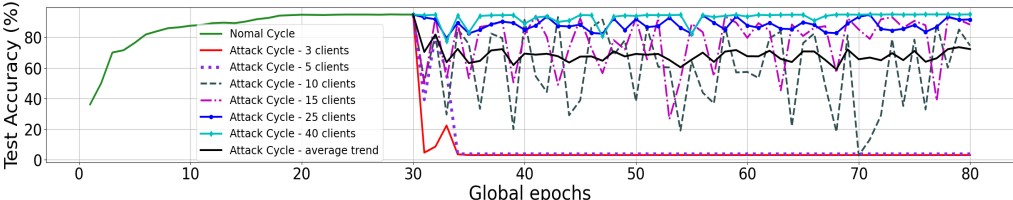

(a) Single-client attack on GTSRB under SimBA-Het attack.

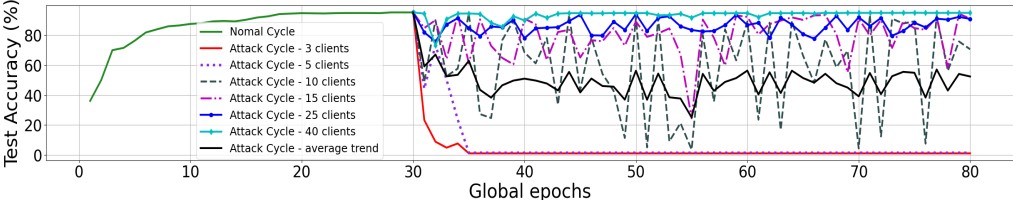

(b) Single-client attack on GTSRB under F-CimBA-Het attack.

Figure 12: Test Accuracy of the normal cycle and single-client attack cycle of proposed F-CimBA framework against SimBA attack along with average attack trend under heterogeneous FL settings.

## 9    Limitations and Future Work

Despite the effectiveness of our F-CimBA, there are certain limitations, which we plan to explore in our future work.

- **Adaptive parameter engineering:** The effectiveness of F-CimBA relies on a key parameter called the noise coefficient ($\mu$), which is initially set to 0.5 in our experiments. In future research, we will explore dynamic parameter learning to potentially enhance F-CimBA's impact by adapting this parameter during training. Recognizing the imperative need for robust attack visibility analysis in the black-box threat model, we plan to investigate alternative metrics and scenarios. This includes examining different threat models, encompassing both full and partial attacker knowledge settings, for a comprehensive understanding of F-CimBA's behaviour.

- **Applicability to other FL tasks:** In future research, we will extend our investigation to object detection and segmentation within the federated learning framework. Benign clients will contribute regular updates for training these models, while malicious clients will employ F-CimBA to poison the training data. The aggregated global model is anticipated to show compromised performance in detecting or segmenting test data. This strategic expansion of our research aims to provide valuable insights into task-specific challenges.

## 10    Conclusion

In this paper, we propose F-CimBA, a feedback-guided causative image black-box attack in federated learning, considering a cautious attacker's perspective. Current attack methods often exhibit substantial attack visibility while retaining impact, attributed to their inherent nature, the scale of perturbations, and a lack of strategies to evade detection mechanisms. Therefore, low attack visibility is essential for maintaining a covert operation, enabling attackers to accomplish their goals without activating the server's defense mechanisms. Consequently, the core of F-CimBA lies in reducing the loss linked to the most confused class, leveraging its probability as feedback to guide the search and refinement of adversarial gradient perturbations. This approach helps to exploit local model vulnerabilities, ensure early attack convergence, and achieve higher misclassification rates. Our empirical results demonstrate F-CimBA's effectiveness in poisoning client training data, causing substantial mispredictions of the global model. In addition, F-CimBA exhibits robustness

against Byzantine-robust aggregation techniques at the server. Also, our investigation into system hardware metrics has revealed the adverse impact of our attack on their performance. In our future works, we aim to explore adaptive parameter engineering techniques, focusing on the dynamic learning of the $\mu$ parameter. Additionally, we intend to understand the impact of utilizing both full and partial attacker knowledge settings to enhance the analysis of attack visibility. Furthermore, our research plan includes evaluating the performance of F-CimBA across diverse tasks, such as object detection and object segmentation within the federated learning framework.

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
