# OpenReview forum: "Feedback-Guided Black-box Attack in Federated Learning: A Cautious Attacker Perspective"
_TMLR — Rejected by TMLR_

### Review · Reviewer_L7t8 · 2025-03-14

**Summary Of Contributions:**

This paper proposes a stealthy data poisoning attack called F-CimBA, in which malicious workers randomly perturb local images and then select the adversarial image-label pairs with the highest prediction probability. The authors validate the effectiveness of the proposed F-CimBA attacks empirically.

**Audience:**

Yes

**Broader Impact Concerns:**

There are no concerns on the ethical implications of this work.

**Claims And Evidence:**

No

**Requested Changes:**

My detailed requested changes are listed in the Weakness part; please refer to it.

**Strengths And Weaknesses:**

**Strengths**

1. The proposed F-CimBA attacks are effective against several existing defense methods, such as Krum, Trimmed-Mean, and FLDefender.

**Weaknesses**

1. The writing of this paper needs improvement. Many terms are not defined, making this paper difficult to follow. For example, what do “B” and “W” represent in Table 1? Moreover, what is the definition of $\max_{\tilde{\mathcal{Y}}_s \neq \mathcal{Y}_s} \{P(\tilde{\mathcal{Y}}_s | \tilde{\mathcal{X}}_s)\}$ in Lines 10 and 16 of Algorithm 1, since $\tilde{\mathcal{Y}}_s$ is a given constant. Additionally, what exactly is the FFA method mentioned in Theorem 5.1, and how is “$\delta$-bounded impact” defined?

2. The theoretical results in Theorem 5.1 are unclear. If I understand correctly, Theorem 5.1 is intended to demonstrate the stealthiness of the proposed attack. However, the upper bound $\delta$ could be large—or even unbounded—in which case the attack would no longer be stealthy. Moreover, the proof of Theorem 5.1 is not sufficiently clear; the authors only present the key inequality without adequately explaining its origin. I suggest that the authors revise and clarify the proof.

3. In the experiments, the authors use different settings for different datasets. For instance, they use a full participation setting for the GTSRB and KBTS datasets, but a partial participation setting for the CIFAR10 and EMNIST datasets. I recommend that the authors adopt a consistent setting across all datasets to facilitate fair comparison.

4. The set of defenses considered in this paper is incomplete, as it omits several important state-of-the-art defenses such as FABA [1] and centered clipping [2]. I suggest that the authors also evaluate the performance of the proposed attack against these defenses.

    [1] Qi Xia, Zeyi Tao, Zijiang Hao, and Qun Li. FABA: an algorithm for fast aggregation against Byzantine attacks in distributed neural networks. In International Joint Conferences on Artificial Intelligence, 2019.

    [2] Sai Praneeth Karimireddy, Lie He, and Martin Jaggi. Learning from history for Byzantine robust optimization. In International Conference on Machine Learning, pages 5311–5319, 2021.

---

> ### Author Response · Authors · 2025-06-10
> **2. Response to the concern related to Theorem 5.1**
>
> Thank you for highlighting the issue regarding the lack of rigour in our original definition of $\delta$. In response, we revised Theorem 5.1 by introducing standard Lipschitz continuity assumptions and derive a tighter and justified bound: $\delta = \alpha \beta \mu$, which captures the actual impact of $\mu$-bounded adversarial updates on test loss.
>
> **Corollary 1**
> Let $\mathcal{G}\_{\theta\_g}$ be the global model with parameters $\theta\_g$ trained in a federated setting over $n$ clients, among which $m$ clients are malicious. Let each local model $f\_{\theta}$ be $\beta$-Lipschitz and the loss function $\mathcal{L}$ be $\alpha$-Lipschitz with respect to its input. Suppose $\mathcal{A}$ is an attacker that perturbs each local input $\mathcal{X}\_i$ to a poisoned input $\tilde{\mathcal{X}}\_i$ using $\mu$-bounded perturbation, i.e., $\|\tilde{\mathcal{X}}\_i - \mathcal{X}\_i\|\_2 \leq \mu$ [1].
>
> Then for each poisoned sample $(\tilde{\mathcal{X}}\_i, \mathcal{Y}\_i)$ generated by an adversarial client $k$, the increase in per-sample loss is bounded as:
> $|\mathcal{L}(f\_{\theta}(\tilde{\mathcal{X}}\_i), \mathcal{Y}\_i) - \mathcal{L}(f\_{\theta}(\mathcal{X}\_i), \mathcal{Y}\_i)| \leq \alpha \beta \mu.$
>
>
> **Theorem 5.1** (**Convergence of F-CimBA under $\mu$-bounded adversarial perturbations**)
>
> Consider a federated learning system with $n$ clients, out of which $m$ are malicious ($m \ll n$). Let $\mathcal{G}\_{\theta\_g}$ denote the global model and $f\_{\theta}$ denote the local model used by client $k$. During round $t$, a malicious client $\mathcal{A}$ generates $\mu$-bounded adversarial samples $\tilde{\mathcal{D}}\_k$ by applying a random gradient perturbation $\mathcal{U}$ to the clean inputs such that $\| \tilde{\mathcal{X}}\_i - \mathcal{X}\_i \|\_2 \leq \mu$.
>
> Let $\tilde{\theta}\_k^t$ represent the poisoned model update from the malicious client $k$, and assume that $f\_{\theta}$ is $\beta$-Lipschitz and $\mathcal{L}$ is $\alpha$-Lipschitz with respect to model output. Then, the average test loss of the global model over $T$ communication rounds satisfies the following bounded convergence criterion:
> $\frac{1}{T} \sum\_{t=1}^{T} \left[ \mathcal{L}(\theta\_g^t + \lambda\_k \tilde{\theta}\_k^t, \mathcal{D}\_{test}) - \mathcal{L}(\theta\_g^t, \mathcal{D}\_{test}) \right] \leq \delta,$
>
> where $\delta = \alpha \beta \mu$.
>
> **Proof Sketch**: The proof of this theorem relies on Corollary 1, which establishes that if adversarial perturbations to local training data are $\mu$-bounded in $L\_2$ norm, and both the model and loss function satisfy Lipschitz continuity assumptions, then the resulting increase in the global test loss is bounded by $\alpha \beta \mu$. This confirms the convergence behavior of the global model under repeated adversarial updates from F-CimBA.
>
>
> **Proof**
> We begin by considering the poisoned input $\tilde{\mathcal{X}}\_i$ and its clean counterpart $\mathcal{X}\_i$ from a malicious client $\mathcal{A}$ such that $\|\tilde{\mathcal{X}}\_i - \mathcal{X}\_i\|\_2 \leq \mu$.
>
> **Step 1: Bounding model output change due to input perturbation.**
>
> Since the local model $f\_{\theta}$ is $\beta$-Lipschitz, we have:
> $\|f\_{\theta}(\tilde{\mathcal{X}}\_i) - f\_{\theta}(\mathcal{X}\_i)\|\_2 \leq \beta \|\tilde{\mathcal{X}}\_i - \mathcal{X}\_i\|\_2 \leq \beta \mu.$
>
> **Step 2: Bounding loss change due to model output shift.**
>
> Because the cross-entropy loss $\mathcal{L}$ is $\alpha$-Lipschitz with respect to its input (e.g., predicted probability vector), we get:
> $|\mathcal{L}(f\_{\theta}(\tilde{\mathcal{X}}\_i), \mathcal{Y}\_i) - \mathcal{L}(f\_{\theta}(\mathcal{X}\_i), \mathcal{Y}\_i)| \leq \alpha \cdot \|f\_{\theta}(\tilde{\mathcal{X}}\_i) - f\_{\theta}(\mathcal{X}\_i)\|\_2.$
>
> Substituting the previous bound:
> $|\mathcal{L}(f\_{\theta}(\tilde{\mathcal{X}}\_i), \mathcal{Y}\_i) - \mathcal{L}(f\_{\theta}(\mathcal{X}\_i), \mathcal{Y}\_i)| \leq \alpha \beta \mu.$
>
> **Step 3: Aggregated loss impact in Federated Learning.**
>
> In FL, the adversarial client $k$ computes poisoned model updates $\tilde{\theta}\_k^t$ based on poisoned local training. These are aggregated with benign client updates by the server using the FedAvg rule:
> $\theta\_g^{t+1} = \sum\_{i=1}^n \lambda\_i \theta\_i^t = \lambda\_k \tilde{\theta}\_k^t + \sum\_{i \neq k} \lambda\_i \theta\_i^t.$
>
> The global model now carries adversarial influence $\lambda\_k \tilde{\theta}\_k^t$ in its update.
>
> Over $T$ global epochs, we compute the average increase in the loss over the test data $\mathcal{D}\_{test}$ caused by this adversarial update:
> $\frac{1}{T} \sum\_{t=1}^{T} \left[ \mathcal{L}(\theta\_g^t + \lambda\_k \tilde{\theta}\_k^t, \mathcal{D}\_{test}) - \mathcal{L}(\theta\_g^t, \mathcal{D}\_{test}) \right] \leq \delta,$
> where each per-round deviation is bounded above by $\alpha \beta \mu$, yielding $\delta = \alpha \beta \mu$.
>
> [1] Li et al. "**Data valuation and detections in federated learning**," in *CVPR* 2024.

---

> ### Author Response · Authors · 2025-06-11
> **1. Response to concern related to clarity and completeness of the definitions**
>
> Thank you for your feedback regarding the clarity and completeness of definitions in the manuscript. We acknowledge the oversight and have made the following clarifications:
>
> 1. In Table 1, “B” and “W” denote black-box and white-box attack settings, respectively. Specifically, "B" refers to a setting where the adversary generates malicious data with limited or no access to model internals, while "W" assumes access to the model’s parameters. We will mention this in our revised manuscript.
>
>
> 2. In Algorithm 1, $\tilde{\mathcal{Y}}\_s$ is not a fixed constant. It refers to the predicted label for the perturbed input $\tilde{\mathcal{X}}\_s$ at each iteration. This value changes dynamically as $\tilde{\mathcal{X}}\_s$ is updated.
> The feedback expression $\max_{\tilde{\mathcal{Y}}_s \neq \mathcal{Y}_s} {P(\tilde{\mathcal{Y}}_s|\tilde{\mathcal{X}}_s)}$ calculates the confidence score of the most probable incorrect class at that point. This directly aligns with the formulation in Page 8, line 10 of the manuscript and is used to guide the adversarial update procedure.
>
>
> 3. The reference to “FFA” in Theorem 5.1 was a typographical error. It should correctly refer to our proposed method, F-CimBA. We have corrected this in the revised version.
>
> 4. Regarding the term “$\delta$-bounded impact”: we have formally defined this in the revised Theorem 5.1. By introducing standard Lipschitz continuity assumptions (the model is $\beta$-Lipschitz and the loss function is $\alpha$-Lipschitz), we now rigorously derive that the change in test loss due to $\mu$-bounded adversarial perturbations is bounded by $\delta = \alpha \beta \mu$ (Please refer to Response 2). This provides a principled justification for the convergence claim and resolves the ambiguity in the earlier version.
>
>
> These revisions aim to improve clarity and ensure the theoretical claims are well-justified. We appreciate your attention to detail. We hope this clarifies both the intent and mechanism of the F-CimBA perturbation loop and will ensure that these points are more explicitly stated in the updated manuscript.

---

> ### Author Response · Authors · 2025-06-16
> **4. Response to evaluation against FABA and centered clipping defenses**
>
> We thank you for pointing out the omission of additional robust aggregation defenses. In response, we conducted new experiments incorporating the suggested FABA and Centered Clipping methods. We evaluated them against our proposed F-CimBA attack on the GTSRB dataset, using 40 clients, all selected for aggregation under a single-client attack scenario. These defenses were compared alongside Krum, LoMar, and FLDefender, which were already included in the main paper, under multiple attacks: DPA-DLF, SimBA, LIE, and F-CimBA. Below are the results:
>
>
> | Defense Method      | DPA-DLF         | SimBA           | LIE             | F-CimBA (Ours)   |
> |---------------------|------------------|------------------|------------------|------------------|
> | No Defense          | 80.08 ± 1.28     | 80.98 ± 1.32     | 79.36 ± 1.48     | **78.43 ± 1.42** |
> | Krum                | 79.45 ± 1.27     | 79.91 ± 0.98     | 80.24 ± 0.12     | **73.95 ± 1.41** |
> | LoMar               | 80.45 ± 2.30     | 81.16 ± 1.87     | 81.13 ± 1.59     | **75.12 ± 0.39** |
> | FLDefender          | 82.34 ± 0.57     | 80.23 ± 1.29     | 82.16 ± 0.81     | **74.36 ± 1.07** |
> | FABA                | 83.73 ± 0.27     | 82.53 ± 1.24     | 83.36 ± 1.68     | **80.08 ± 0.87** |
> | Centered Clipping   | 86.27 ± 1.03     | 85.97 ± 1.35     | 85.24 ± 1.38     | **81.29 ± 1.93** |
>
> **Key Observations:**
> * F-CimBA remains consistently more effective in degrading the global model accuracy across all defense mechanisms compared to other attacks.
>
> * Even under strong defenses like FABA and Centered Clipping, which show higher robustness across DPA, SimBA, and LIE, F-CimBA is still able to reduce the test accuracy to 80.08% and 81.29%, respectively, showing that our attack can partially bypass these advanced defenses.
>
> * Under Krum, LoMar, and FLDefender, F-CimBA achieves the lowest accuracy among all attacks, with a drop of up to 4.5–6% more than the next best attack, further demonstrating its stealth and effectiveness.
>
> * Notably, F-CimBA degrades performance more than SimBA and LIE under every defense method tested, showing its superiority as a black-box, low-visibility attack strategy.
>
> These results confirm that F-CimBA remains effective even against various defense methods, and we will include these results and observations in the revised manuscript. We hope this addresses your concern.

---

> ### Author Response · Authors · 2025-06-17
> **3. Response to the concern on the consistency of participation settings across datasets**
>
> We thank you for the insightful suggestion. To ensure fairness and consistency in evaluation, we conducted additional experiments on the GTSRB dataset under a partial participation setting, where 40 clients were simulated, and 20 clients were randomly selected per round for aggregation. The setting uses a single-client attacker, and the same defense baselines used in the paper were applied, including Krum, LoMar, FLDefender, the suggested FABA, and Centered Clipping, evaluated under different attack methods.
>
> The results are given below
>
> | Defense Method      | DPA-DLF         | SimBA           | LIE             | F-CimBA (Ours)   |
> |---------------------|------------------|------------------|------------------|------------------|
> | No Defense          | 82.46 ± 1.31     | 81.86 ± 0.87     | 78.90 ± 1.68     | **77.54 ± 1.42** |
> | Krum                | 78.47 ± 1.05     | 80.38 ± 0.73     | 81.36 ± 1.24     | **72.12 ± 1.17** |
> | LoMar               | 81.78 ± 0.94     | 82.34 ± 0.86     | 80.15 ± 0.98     | **74.68 ± 1.63** |
> | FLDefender          | 81.38 ± 0.91     | 81.68 ± 1.09     | 83.11 ± 0.72     | **76.27 ± 1.11** |
> | FABA                | 84.30 ± 0.84     | 83.76 ± 1.14     | 84.90 ± 0.99     | **81.21 ± 1.22** |
> | Centered Clipping   | 85.81 ± 1.09     | 84.01 ± 1.37     | 83.71 ± 1.48     | **80.89 ± 0.96** |
>
> **Key Observations:**
> * F-CimBA continues to exhibit the highest degradation in test accuracy across all defenses, even under partial client participation for the GTSRB dataset, confirming its robustness and generalizability.
>
> * Under Krum, F-CimBA reduces accuracy to 72.12%, which is 6.35–9.24% lower than competing attacks in the same setting.
>
> * Even under strong defenses such as FABA and Centered Clipping, F-CimBA achieves 81.21% and 80.89%, respectively, still outperforming all other attack baselines in terms of degradation, similar to results on other datasets like CIFAR10 and EMNIST.
>
> * These results are consistent with those obtained under full participation of GTSRB, confirming that our earlier choice of using both full and partial participation across datasets was a reasonable trade-off to avoid redundancy in the results and observations and maintain brevity in the main paper.
>
> We will include these new results and note the rationale for mixed participation settings in the revised manuscript. We hope this effectively addresses your concern.

---

### Review · Reviewer_DVYf · 2025-04-13

**Summary Of Contributions:**

1. The paper introduces F-CIMBA, a black-box data poisoning adversarial attack method designed for federated learning environments. It employs a gradient-based perturbation strategy to poison client data, which is then used to train local client models, turning them into malicious clients. These clients contribute harmful gradient updates to the global model, thereby indirectly degrading its overall performance.
2. F-CIMBA is novel in that it operates under black-box constraints: it does not modify the training process, alter the server’s aggregation strategy, or interfere with server communications, all while being invisible.
3. The paper includes extensive ablation studies and experiments to validate the effectiveness of the proposed attack.
4. Additional experiments are conducted to evaluate the method against various defense strategies, as well as analyses of its impact under non-IID data conditions and on hardware-related metrics.
5. The paper is well-written and easy to understand.

**Audience:**

Yes

**Broader Impact Concerns:**

No major concerns.

**Claims And Evidence:**

Yes

**Requested Changes:**

See above.

**Strengths And Weaknesses:**

1. It is not clear how is the perturbation added at each step. In section 4, page 8 it is mentioned that.
`For the initial iteration, the random gradient perturbation (U) is added in the positive direction. The gradient is added in the negative direction for further iterations and is changed to random perturbations in the subsequent iterations.`
It is not clear how is the gradient computed. Is the gradient based on the loss of the most confused class w.r.t the perturbed input image? If this is the case than F-CIMBA is not fully black box as it requires the gradient of the local models to generate perturbations.
2. It is hard to understand why in federated learning setting, malicious gradient updates will **always** indirectly poision the global model thus degrading its performance. The proof in Section-5 develops a relationship between µ (noise coefficient) and the maximum increase in loss. However, if µ is low (to ensure attack is invisible), the loss increase will be low as well, which does not explain why the proposed method should always work.
3. It will be good to experiment with various model architectures like ViTs, to further demonstrate the effectiveness of F-CIMBA.

---

> ### Author Response · Authors · 2025-06-10
> **1. Clarification on perturbation process, black-box setting, and gradient use in F-CimBA**
>
> Thank you for your thoughtful comment. We understand the confusion arising from the phrase "random gradient perturbation" in Section 4 and appreciate the opportunity to clarify the perturbation mechanism in our F-CimBA attack.
>
> F-CimBA is designed for black-box adversarial data poisoning, where gradient information is not accessible. As correctly pointed out on Page 8, line 3 of our manuscript:
>
> "Since the gradient information is not available for the black-box model $f\_\theta$, the output probabilities act as an effective proxy to direct the search for the perturbation that generates the adversarial image. In this untargeted attack scenario, the goal is to increase the probability of the most confused class."
>
> To implement this, we do not compute or estimate gradients explicitly. Instead, we adopt a feedback-driven, uniform noise-based perturbation strategy. As described in Page 8, line 7,
>
> "...the F-CimBA adversarial client generates a random gradient perturbation $\mathcal{U}$, which is then scaled by a noise coefficient $\mu$, such that the prediction $f\_\theta(\mathcal{X}\_i + \mu \times \mathcal{U})$ does not match the label $\mathcal{Y}\_i$..."
>
> We acknowledge that the term "random gradient perturbation" is misleading in a black-box context. What we meant is random perturbation noise $\mathcal{U}$, sampled from a uniform distribution, used to explore directions in the input space. We will rewrite our text to address this concern and avoid misinterpretation.
>
> Our approach aligns with prior black-box attack strategies, such as Bandits with Priors [1], which performs gradient-free query-based optimization. This approach allows adversaries to estimate gradient-like information without access to true model gradients. In our case, instead of constructing full finite-difference gradient estimates (which can be query-expensive in high dimensions), we use randomly sampled directions $\mathcal{U}$ and rely on the feedback signal from the model's prediction probabilities to guide iterative search, thus remaining within practical black-box constraints.
>
> To further clarify, the mention of "positive" and "negative" directions refers to the sign of the perturbation applied in each iteration relative to the prior feedback and not to any true gradient-derived direction. Importantly, this avoids explicit gradient computation and makes F-CimBA both lightweight and compatible with the strict black-box threat model, where only output logits or class probabilities can be queried.
>
> We will revise the relevant section to replace the phrase "random gradient perturbation" with "random perturbation noise" and better clarify this iterative feedback-based search process. We thank you for highlighting this point, which will help improve the clarity and correctness of the paper. We hope this addresses the concern and clarifies the confusion regarding our perturbation mechanism and black-box setting.
>
> [1]  Ilyas et al. "**Prior Convictions: Black-box Adversarial Attacks with Bandits and Priors**," in *ICLR* 2019

---

> ### Author Response · Authors · 2025-06-13
> **2. Justification for the effectiveness of malicious updates and F-CimBA low-visibility perturbations in FL**
>
> Thank you for raising these critical points.
>
> 1. **Reasoning on why malicious gradient updates in FL tend to poison the global model.** We agree that it is crucial to justify why, in the FL setting, malicious gradient updates will always indirectly poison the global model, thus degrading its performance.
> The fundamental reason lies in how standard FL systems aggregate client updates. Under standard aggregation schemes like FedAvg, well-designed malicious updates, especially those that remain undetected, **consistently bias the global model over time, leading to degradation in performance**. This effect has been **both theoretically analyzed and empirically validated in multiple prior works [1][2][3]**. When a malicious client submits an adversarial update, crafted through either data poisoning or model poisoning, it contributes directly to the global model update, suboptimal or adversarially biased decision boundaries that reduce test accuracy, induce targeted misbehaviour or slow convergence. The server aggregates local model updates in FL, assuming that all clients act honestly. This lack of verification creates an inherent vulnerability, as shown in prior works below.
> * *Shejwalkar et al. [1]* comprehensively evaluate untargeted poisoning attacks and confirm that malicious updates can successfully bias the global model when not effectively filtered by the defense, even under constrained adversarial capabilities.
> * *Usynin et al. [2]* highlight that adversarial interference in collaborative learning leads to meaningful reductions in utility, even when perturbations are subtle. They show that even privacy-preserving FL systems are vulnerable to low-visibility attacks that do not require gradient access yet can significantly undermine the utility of the global model over multiple communication rounds.
> * *Kumar et al. [3]* explain that attacks with low attack visibility and small budgets can still have a high impact, particularly when the global model incorporates updates without sufficient scrutiny. These insights reinforce that well-designed poisoning attacks do not need to be large in magnitude to be impactful.
>
>
> 2. **Regarding the reason on why small $\mu$ is still effective in F-CimBA.** It is indeed true that a smaller value of $\mu$ bounds the perturbation magnitude and, in theory, limits the per-sample increase in loss. However, several key factors explain why even small $\mu$ values can still lead to effective and sustained model degradation in our F-CimBA attack
> * *FL accumulates small deviations over rounds.*  FL is an iterative process where client updates are aggregated repeatedly across communication rounds. Even if the per-update impact of a low-$\mu$ perturbation is small, its cumulative effect over $T$ rounds becomes significant, particularly when the adversarial client participates repeatedly (online attack). This compounding nature of poisoning, also highlighted in prior work such as Usynin et al. [2], allows small, undetectable updates to gradually shift the global model toward an adversarial objective.
> * *Our attack focuses on model confusion, not large-scale loss spikes.* F-CimBA does not aim to maximise loss arbitrarily but rather to increase the confidence of incorrect class predictions (i.e., the most confused class). This strategy does not require large perturbations. Instead, it takes advantage of **local model weaknesses and decision boundaries**, which can be nudged effectively with small perturbations.  Using a small $\mu$ ensures the perturbed samples stay close to the original data distribution, making them less likely to be detected or filtered by defenses. The result is a more stealthy degradation of performance, which is visible in our empirical results.
> * *Empirical evidence shows clear impact even at low $\mu$.* With $\mu = 0.5$ (used in our experiments), we achieve a test accuracy degradation of up to 11% without triggering defenses. This validates that small $\mu$ can still achieve strong practical impact due to the attack's cumulative and focused nature.
>
> We hope this clarifies that, in practical FL settings, even low-visibility malicious updates, when aggregated, can consistently poison the global model and degrade its performance. While smaller \$\mu\$ limits the immediate effect, F-CimBA leverages iteration, subtle misdirection, and low detectability to reliably cause degradation over time. We will incorporate these explanations into the revised manuscript for clarity.
>
> [1] Shejwalkar et al. "**Back to the drawing board: A critical evaluation of poisoning attacks on production federated learning**," in *IEEE S&P*, 2022.
>
> [2] Usynin et al. "**Adversarial interference and its mitigations in privacy-preserving collaborative machine learning,**" in *Nature Machine Intelligence*, 2021.
>
> [3] Kumar et al. "**The impact of adversarial attacks on federated learning: A survey,**" in *IEEE TPAMI*, 2024.

---

> ### Author Response · Authors · 2025-06-16
> **3. Response on model architecture diversity and effectiveness of F-CimBA**
>
> We thank you for the valuable suggestion. To further validate the effectiveness and generalizability of F-CimBA, we conducted additional experiments on the CIFAR-10 dataset using three diverse and widely adopted model architectures: ResNet-50, DenseNet-121, and Vision Transformer (ViT). The federated learning setup was configured with $n = 100$ clients, $k = 40$ clients selected per round, $10$ malicious clients, and a non-IID Dirichlet partitioning with $\beta = 1$ to simulate realistic data heterogeneity. These models were evaluated against multiple poisoning and black-box baselines, including DPA-DLF, DPA-SLF, LIE, Fang et al., and SimBA, as well as a no-attack reference. The results are summarized below:
>
>
> | Architecture         | No Attack       | DPA-DLF         | DPA-SLF         | LIE             | Fang et al.     | SimBA           | F-CimBA (Ours)   |
> | -------------------- | --------------- | --------------- | --------------- | ----------------| ----------------| ----------------| ---------------- |
> | ResNet-50            | 94.12 ± 1.26    | 93.57 ± 0.64    | 93.02 ± 1.41    | 91.37 ± 0.92     | 87.48 ± 0.47     | 88.36 ± 1.84     | **85.67 ± 1.03** |
> | DenseNet-121         | 92.64 ± 0.83    | 91.95 ± 1.12    | 91.57 ± 0.76    | 90.83 ± 1.67     | 90.36 ± 1.49     | 90.74 ± 0.94     | **88.31 ± 0.65** |
> | Vision Transformer   | 95.85 ± 1.38    | 94.45 ± 0.79    | 94.56 ± 1.42    | 93.27 ± 0.93     | 92.34 ± 1.51     | 92.01 ± 0.68     | **91.44 ± 1.23** |
>
>
> **Key Observations:**
> * Across all three model architectures, F-CimBA consistently causes the largest drop in test accuracy, confirming its strong poisoning effect even under different architectural biases and learning dynamics.
>
> * On ResNet-50, F-CimBA lowers the accuracy to 85.67%, which is over 8.4% lower than the clean model and 1.8–5.7% lower than competing attacks.
>
> * On DenseNet-121, it reduces performance to 88.31%, maintaining at least a 2.0–3.6% gap over other attacks, including LIE and SimBA.
>
> * On ViT, a transformer-based model with significantly different internal mechanisms, F-CimBA still causes the most performance degradation (91.44%) compared to DPA, LIE, and Fang et al., indicating that it is effective even in architectures designed for better robustness.
>
> These results reinforce the architecture-agnostic nature of F-CimBA, demonstrating its ability to successfully degrade performance across convolutional and transformer-based models. We will include these findings in the revised manuscript to strengthen our empirical validation and to directly address the suggestion. We hope this addresses the concern.

---

### Review · Reviewer_e2Zf · 2025-06-07

**Summary Of Contributions:**

This paper proposes a data poisoning attack in the federated learning setting. In particular, the attacker has access to the data, but not to the local or global models. The attacker's aim is to add perturbations to the training data (only $x$, not the labels $y$) such that the perturbed data minimizes the loss on the label with the highest softmax score for an incorrect output with respect to the current state of the global model. This poisoned data (along with an adjusted label corresponding to the incorrect output) is then used to compute the current update for the client that is being attacked. Since the global model is now learning to associate minimally modified inputs from a given class with the incorrect class label, the attack leads to a reduced overall test accuracy. This is validated with extensive experiments over 4 computer vision datasets with both iid and non-iid distribution of data across clients. A stealth metric comparing the validation accuracy of the local and global models is also used to monitor undetectability.

**Audience:**

Yes

**Broader Impact Concerns:**

This paper proposes attacking existing distributed learning algorithms. If used incorrectly, these attacks can have a negative impact on deployed machine learning systems so this should be addressed in the broader impact statement.

**Claims And Evidence:**

No

**Requested Changes:**

-> Check the proof of Theorem 5.1. and add the assumptions required for the claims there to hold. I believe at least something like a Lipschitz condition is required.
-> How is the random perturbation noise generated? Please clarify and add a detailed algorithm.

**Strengths And Weaknesses:**

*Strengths*
+ Extensive experimental evaluation using different datasets, aggregation algorithms, as well as comparisons to existing work

*Weaknesses*
- The proof of Theorem 5.1. is vacuous, and possibly just incorrect. The penultimate equation on page 10 is invalid. It directly relates the loss on a perturbed model to that for an unperturbed model and upper bounds it by $\mu$. However, $\mu$ is a bound on the perturbation added to poisoned data. Without additional strong assumptions on the loss function, it is unclear how that equation is valid.
- In Algorithm 1, the most crucial step is perhaps defining how the random perturbation noise is generated using the feedback from the model in order to increase the classification probability of the incorrect class. However, this description is entirely missing from the paper. It is unclear how to evaluate the technical correctness without this information.
- Table 9 refers to a 1A attack setting but this is never defined. In addition, it appears the GTA is lower with Krum than without it (comparing Table 9 and Table 6 for the GTSRB dataset), which seems odd but is not explained further

---

> ### Author Response · Authors · 2025-06-10
> **1. Response to the concern related to Theorem 5.1**
>
> Thank you for your insightful comment on Theorem 5.1. We agree that the original formulation lacked a formal justification and we have revised the theorem by assuming that the local model $f_{\theta}$ is $\beta$-Lipschitz and the loss function $\mathcal{L}$ is $\alpha$-Lipschitz. We believe this strengthens the theoretical foundation of our work and addresses your concern.
>
> **Corollary 1**
> Let $\mathcal{G}\_{\theta\_g}$ be the global model with parameters $\theta\_g$ trained in a federated setting over $n$ clients, among which $m$ clients are malicious. Let each local model $f\_{\theta}$ be $\beta$-Lipschitz and the loss function $\mathcal{L}$ be $\alpha$-Lipschitz with respect to its input. Suppose $\mathcal{A}$ is an attacker that perturbs each local input $\mathcal{X}\_i$ to a poisoned input $\tilde{\mathcal{X}}\_i$ using $\mu$-bounded perturbation, i.e., $\|\tilde{\mathcal{X}}\_i - \mathcal{X}\_i\|\_2 \leq \mu$ [1].
>
> Then for each poisoned sample $(\tilde{\mathcal{X}}\_i, \mathcal{Y}\_i)$ generated by an adversarial client $k$, the increase in per-sample loss is bounded as:
> $|\mathcal{L}(f\_{\theta}(\tilde{\mathcal{X}}\_i), \mathcal{Y}\_i) - \mathcal{L}(f\_{\theta}(\mathcal{X}\_i), \mathcal{Y}\_i)| \leq \alpha \beta \mu.$
>
> **Theorem 5.1** (**Convergence of F-CimBA under $\mu$-bounded adversarial perturbations**)
>
> Consider a federated learning system with $n$ clients, out of which $m$ are malicious ($m \ll n$). Let $\mathcal{G}\_{\theta\_g}$ denote the global model and $f\_{\theta}$ denote the local model used by client $k$. During round $t$, a malicious client $\mathcal{A}$ generates $\mu$-bounded adversarial samples $\tilde{\mathcal{D}}\_k$ by applying a random gradient perturbation $\mathcal{U}$ to the clean inputs such that $\| \tilde{\mathcal{X}}\_i - \mathcal{X}\_i \|\_2 \leq \mu$.
>
> Let $\tilde{\theta}\_k^t$ represent the poisoned model update from the malicious client $k$, and assume that $f\_{\theta}$ is $\beta$-Lipschitz and $\mathcal{L}$ is $\alpha$-Lipschitz with respect to model output. Then, the average test loss of the global model over $T$ communication rounds satisfies the following bounded convergence criterion:
> $\frac{1}{T} \sum\_{t=1}^{T} \left[ \mathcal{L}(\theta\_g^t + \lambda\_k \tilde{\theta}\_k^t, \mathcal{D}\_{test}) - \mathcal{L}(\theta\_g^t, \mathcal{D}\_{test}) \right] \leq \delta,$
>
> where $\delta = \alpha \beta \mu$.
>
> **Proof Sketch**: The proof of this theorem relies on Corollary 1, which establishes that if adversarial perturbations to local training data are $\mu$-bounded in $L\_2$ norm, and both the model and loss function satisfy Lipschitz continuity assumptions, then the resulting increase in the global test loss is bounded by $\alpha \beta \mu$. This confirms the convergence behavior of the global model under repeated adversarial updates from F-CimBA.
>
>
> **Proof**
> We begin by considering the poisoned input $\tilde{\mathcal{X}}\_i$ and its clean counterpart $\mathcal{X}\_i$ from a malicious client $\mathcal{A}$ such that $\|\tilde{\mathcal{X}}\_i - \mathcal{X}\_i\|\_2 \leq \mu$.
>
> **Step 1: Bounding model output change due to input perturbation.**
>
> Since the local model $f\_{\theta}$ is $\beta$-Lipschitz, we have:
> $\|f\_{\theta}(\tilde{\mathcal{X}}\_i) - f\_{\theta}(\mathcal{X}\_i)\|\_2 \leq \beta \|\tilde{\mathcal{X}}\_i - \mathcal{X}\_i\|\_2 \leq \beta \mu.$
>
> **Step 2: Bounding loss change due to model output shift.**
>
> Because the cross-entropy loss $\mathcal{L}$ is $\alpha$-Lipschitz with respect to its input (e.g., predicted probability vector), we get:
> $|\mathcal{L}(f\_{\theta}(\tilde{\mathcal{X}}\_i), \mathcal{Y}\_i) - \mathcal{L}(f\_{\theta}(\mathcal{X}\_i), \mathcal{Y}\_i)| \leq \alpha \cdot \|f\_{\theta}(\tilde{\mathcal{X}}\_i) - f\_{\theta}(\mathcal{X}\_i)\|\_2.$
>
> Substituting the previous bound:
> $|\mathcal{L}(f\_{\theta}(\tilde{\mathcal{X}}\_i), \mathcal{Y}\_i) - \mathcal{L}(f\_{\theta}(\mathcal{X}\_i), \mathcal{Y}\_i)| \leq \alpha \beta \mu.$
>
> **Step 3: Aggregated loss impact in Federated Learning.**
>
> In FL, the adversarial client $k$ computes poisoned model updates $\tilde{\theta}\_k^t$ based on poisoned local training. These are aggregated with benign client updates by the server using the FedAvg rule:
> $\theta\_g^{t+1} = \sum\_{i=1}^n \lambda\_i \theta\_i^t = \lambda\_k \tilde{\theta}\_k^t + \sum\_{i \neq k} \lambda\_i \theta\_i^t.$
>
> The global model now carries adversarial influence $\lambda\_k \tilde{\theta}\_k^t$ in its update.
>
> Over $T$ global epochs, we compute the average increase in the loss over the test data $\mathcal{D}\_{test}$ caused by this adversarial update:
> $\frac{1}{T} \sum\_{t=1}^{T} \left[ \mathcal{L}(\theta\_g^t + \lambda\_k \tilde{\theta}\_k^t, \mathcal{D}\_{test}) - \mathcal{L}(\theta\_g^t, \mathcal{D}\_{test}) \right] \leq \delta,$
> where each per-round deviation is bounded above by $\alpha \beta \mu$, yielding $\delta = \alpha \beta \mu$.
>
> [1] Li et al. "**Data valuation and detections in federated learning**,"  in *CVPR* 2024.

---

> > ### Comment · Reviewer_e2Zf · 2025-08-04
> > **Proof still has issues**
> >
> > The final statement that is written in the proof above, where the loss of the model at the perturbed parameter vector is bounded above from the loss at the clean parameter vector is incorrect. The loss is directly bounded by the change in the loss output resulting from change in input and Lipschitz assumptions. However, what has changed here is not the input, but the parameter vector *learned* as a consequence of the input changing. Without an analysis of the learning dynamics with poisoned inputs, it is unclear where the poisoned parameter vector is going to lie in parameter space and how it impacts the loss. I recommend that the authors revisit the proof for correctness.

---

> > > ### Author Response · Authors · 2025-08-04
> > > **Response to - Proof still has issues**
> > >
> > > We thank the reviewer for the valuable and insightful comment on our earlier version of Theorem 5.1. We now understand and acknowledge that our earlier argument incorrectly attempted to upper bound the loss at the poisoned model parameters directly using the perturbation bound $\mu$ on the input, without accounting for how this affects the model parameter vector during learning. As pointed out, since it is the parameter vector that changes due to poisoned inputs, bounding the loss requires either an analysis of training dynamics or a control over the parameter shift. Following this direction, and inspired by the treatment in prior works (e.g.,Theorem 2 [1]), we have revised our proof by
> > >
> > > * Introducing explicit Lipschitz continuity assumptions on the model and loss.
> > > * Bounding the change in local model updates due to bounded input perturbations.
> > > * Quantifying how the global model deviates under FedAvg when a subset of clients are adversarial.
> > > * Bounding the resulting change in global loss using standard stability assumptions.
> > >
> > > Our revised proof formalizes how bounded perturbations to input data result in bounded deviation of model updates, which in turn causes a controlled degradation in global model performance under FedAvg. This version avoids the earlier misstep of directly bounding loss through input perturbation and tackles the case. We hope it addresses the raised concern.
> > >
> > > **Revised Theorem 5.1: Convergence of F-CimBA under $\mu$-bounded perturbations**
> > >
> > > Let $f\_{\theta\_g^t}$ denote the global model at round $t$ with parameters $\theta\_g^t$ in a federated learning setup with $n$ clients, of which $m$ are adversarial. Assume the following
> > >
> > > * The local model $f\_\theta$ is $\beta$-Lipschitz with respect to the input $x$.
> > > * The loss function $\mathcal{L}$ is $\alpha$-Lipschitz with respect to the model output.
> > > * Each adversarial client perturbs input $x$ to $\tilde{x}$ such that $\|x - \tilde{x}\| \leq \mu$.
> > > * The client update function is $\gamma$-Lipschitz with respect to the input dataset in $\ell\_2$ norm.
> > >
> > > Then, the average increase in global test loss over $T$ rounds due to $m$ malicious clients performing $\mu$-bounded attacks is upper bounded as:
> > >
> > > $$\frac{1}{T} \sum\_{t=1}^T \left[\mathcal{L}(f\_{\theta\_g^t}, \mathcal{D}\_{\text{test}}) - \mathcal{L}(f\_{\theta\_g^{t,\text{clean}}}, \mathcal{D}\_{\text{test}}) \right] \leq \delta,
> > > $$
> > >
> > > where $\delta = \lambda m \cdot \alpha (\beta + \gamma)\mu$ and $\lambda = \frac{1}{n}$ is the aggregation weight in FedAvg.
> > >
> > > **Proof:** We decompose the impact of poisoned inputs into three effects.
> > >
> > > **Step 1: Effect of Input Perturbation on Model Output**
> > >
> > > For any poisoned input $\tilde{x}$ generated by malicious client $k$
> > > * Since $f\_{\theta}$ is $\beta$-Lipschitz in input
> > > $$
> > > \|f\_\theta(\tilde{x}) - f\_\theta(x)\| \leq \beta \|\tilde{x} - x\| \leq \beta \mu
> > > $$
> > > * Since $\mathcal{L}$ is $\alpha$-Lipschitz in model output
> > > $$
> > > |\mathcal{L}(f\_\theta(\tilde{x}), y) - \mathcal{L}(f\_\theta(x), y)| \leq \alpha \beta \mu
> > > $$
> > > This bounds local loss change per sample due to input perturbation.
> > >
> > > **Step 2: Parameter shift due to poisoned dataset**
> > >
> > > Let $\theta\_k^{t,\text{clean}}$ and $\tilde{\theta}\_k^t$ be the client updates trained on clean and poisoned data respectively. Assume the update function is $\gamma$-Lipschitz w.r.t. data perturbations (supported by Theorem2 [1]):
> > > $$
> > > \|\tilde{\theta}\_k^t - \theta\_k^{t,\text{clean}}\| \leq \gamma \mu
> > > $$
> > > Then the change in model output due to parameter drift
> > > $$\|f\_{\tilde{\theta}\_k^t}(x) - f\_{\theta\_k^{t,\text{clean}}}(x)\| \leq \gamma \mu$$
> > > Hence, loss difference is bounded as:
> > > $$ |\mathcal{L}(f\_{\tilde{\theta}\_k^t}(x), y) - \mathcal{L}(f\_{\theta\_k^{t,\text{clean}}}(x), y)| \leq \alpha \gamma \mu$$
> > >
> > > **Step 3: FedAvg aggregation effect**
> > >
> > > Global model parameters update at round $t$
> > > $$
> > > \theta\_g^t = \sum\_{k =1 }^n \lambda\_k \theta\_k^t = \sum\_{k \in \mathcal{B}} \lambda\_k \theta\_k^t + \sum\_{k \in \mathcal{A}} \lambda\_k \tilde{\theta}\_k^t.
> > > $$
> > >
> > > Define clean global model parameters update (no attack)
> > > $$
> > > \theta\_g^{t,\text{clean}} = \sum\_{k=1}^n \lambda\_k \theta\_k^{t,\text{clean}}
> > > $$
> > > Then, the total global model parameters shift is
> > > $$
> > > \|\theta\_g^t - \theta\_g^{t,\text{clean}}\| \leq \sum\_{k \in \mathcal{A}} \lambda\_k \|\tilde{\theta}\_k^t - \theta\_k^{t,\text{clean}}\| \leq m \cdot \lambda \cdot \gamma \mu
> > > $$
> > > Loss deviation due to global model perturbation
> > > $$
> > > |\mathcal{L}(f\_{\theta\_g^t}, x) - \mathcal{L}(f\_{\theta\_g^{t,\text{clean}}}, x)| \leq \alpha \cdot m \cdot \lambda \cdot \gamma \mu
> > > $$
> > >
> > > **Combining total effects**
> > >
> > > Adding this to the previous loss deviation due to input perturbation ($\alpha\beta\mu$), we get
> > >
> > > $$\delta = \lambda m (\alpha\beta\mu+\alpha\gamma\mu) = \lambda m. \alpha (\beta+\gamma)\mu$$
> > >
> > > Including input-to-output impact and model update impact
> > > $$
> > > \text{Total deviation} \leq \delta
> > > $$
> > >
> > > [1] Li et al. "**Data valuation and detections in federated learning**,"  in *CVPR* 2024.

---

> ### Author Response · Authors · 2025-06-11
> **2. Response to missing description of perturbation noise and feedback mechanism**
>
> We thank you for this valuable observation. In the original version, we acknowledge that the explanation of how random perturbation noise is updated based on feedback was not fully clear. We have now revised **Algorithm 1** to explicitly show that:
>
> 1. The initial perturbation noise $\mathcal{U}$ is sampled from a uniform distribution $\mathcal{U} \sim \text{Uniform}(-1, 1)^{\dim(\mathcal{X}_s)}$, with dimenation same as original source image.
>
> 2. The adversarial sample is created as $\tilde{\mathcal{X}}_s = \mathcal{X}_s + \mu \times \mathcal{U}$.
>
> 3. The feedback is calculated as the maximum predicted probability among incorrect classes: $\text{feedback} = \max_{\tilde{\mathcal{Y}}_s \neq \mathcal{Y}_s} {P(\tilde{\mathcal{Y}}_s \mid \tilde{\mathcal{Y}}_s)}$
>
> 5. If the new feedback does not improve over the previous one:
> * On the first retry, the direction of $\mathcal{U}$ is reversed: $\mathcal{U} \gets -\mathcal{U}$
> * On further retries, a new $\mathcal{U}$ is sampled from the uniform distribution: $\mathcal{U} \sim \text{Uniform}(-1, 1)^{\dim(\mathcal{X}_s)}$
>
> 6. The process repeats until either the feedback improves or the iteration budget is exhausted.
>
> Below is the detailed updated algorithm.
>
> **Inputs:**
> - Global model parameters: $\theta_g^{\text{init}}$
> - Client datasets: $[\mathcal{D}\_k]_{k=1}^n$
> - Noise coefficient: $\mu$
>
> **Output:**
> - Global test accuracy after attack: $\mathcal{A}_\mathcal{G}^*$
>
>
> #### ClientExecution($\mathcal{G}_{\theta_g}^t$)
>
> 1. For each client $k = 1$ to $n$:
>    - **If** ClientType[$k$] == $\mathcal{A}$ (adversarial client):
>      - Initialize local model: $\tilde{\theta}_k^t \gets \theta_g^t$
>      - For each sample $s$ in $\mathcal{D}_k$:
>        - Sample perturbation: $\mathcal{U} \sim \text{Uniform}(-1, 1)^{\dim(\mathcal{X}_s)}$
>        - Compute perturbed input: $\tilde{\mathcal{X}}_s^{(0)} = \mathcal{X}_s + \mu \cdot \mathcal{U}$
>        - Predict: $\tilde{\mathcal{Y}}\_s^{(0)} = f_\theta(\tilde{\mathcal{X}}_s^{(0)})$
>        - Compute feedback:
>          $\text{feedback}\_{\text{old}} = \max_{\tilde{\mathcal{Y}}_s \neq \mathcal{Y}_s} P(\tilde{\mathcal{Y}}_s | \tilde{\mathcal{X}}_s^{(0)})$
>        - Initialize:
>          $\text{feedback}\_{\text{new}} \gets \text{feedback}_{\text{old}}$,
>          iter = max_iterations, $j \gets 0$
>        - **While** $\text{feedback}\_{\text{new}} \leq \text{feedback}_{\text{old}}$ and iter ≠ 0:
>          - **If** $j == 0$:
>               retain $\mathcal{U}$ (positive noise direction)
>          - **Else if** $j == 1$:
>               $\mathcal{U} \gets -\mathcal{U}$ (reverse noise direction)
>          - **Else**:
>               re-sample new $\mathcal{U} \sim \text{Uniform}(-1, 1)^{\dim(\mathcal{X}_s)}$
>          - Update perturbed input:
>            $\tilde{\mathcal{X}}_s^{(j+1)} = \mathcal{X}_s + \mu \cdot \mathcal{U}$
>          - Predict:
>            $\tilde{\mathcal{Y}}\_s^{(j+1)} = f_\theta(\tilde{\mathcal{X}}_s^{(j+1)})$
>          - Update feedback:
>            $\text{feedback}\_{\text{new}} = \max_{\tilde{\mathcal{Y}}_s \neq \mathcal{Y}_s} P(\tilde{\mathcal{Y}}_s | \tilde{\mathcal{X}}_s^{(j+1)})$
>          - iter -= 1, $j \gets j + 1$
>        - Append $(\tilde{\mathcal{X}}_s^{(j)}, \tilde{\mathcal{Y}}_s^{(j)})$ to $\tilde{\mathcal{D}}_k$
>      - Compute poisoned update:
>        $\tilde{\theta}\_k^t \gets \tilde{\theta}\_k^t - \eta \nabla \mathcal{L}(\tilde{f}_{\theta,k}^t(\tilde{\mathcal{D}}_k))$
>    - **Else** (benign client):
>      - Initialize: $\theta_k^t \gets \theta_g^t$
>      - Train on $\mathcal{D}_k$ for $E$ epochs
>      - Update:
>        $\theta_k^t \gets \theta_k^t - \eta \nabla \mathcal{L}(f_{\theta,k}^t(\mathcal{D}_k))$
>      - Scale update:
>        $\theta_k^t \gets \lambda_k \cdot \theta_k^t$
> 2. **Return:**
>    $[\tilde{\theta}\_k^t]\_{k=1}^m \cup [\theta\_k^t]_{k=m+1}^n$
>
>
> #### Main()
>
> 1. Initialize global model: $\theta_g^1 = \theta_g^{\text{init}}$
> 2. For each global epoch $t = 1$ to $T$:
>    - $[\theta\_k^t]\_{k=1}^n = \text{ClientExecution}(\mathcal{G}_{\theta_g}^t)$
>    - Aggregate:
>      $\mathcal{G}_{\theta_g}^{t+1} \gets \text{FedAvg}([\theta\_k^t]\_{k=1}^n)$
>    - Evaluate global model:
>      $\mathcal{A}\_\mathcal{G}^* = \texttt{Test}(\mathcal{G}\_{\theta_g}^{t+1}, \mathcal{D}\_{test})$
> 3. **Return:**
>    $\max(\mathcal{A}_\mathcal{G}^*)$
>
>  It reflects the feedback loop described on Page 8:
> “The adversary uses the local black-box model on $\tilde{\mathcal{X}}_i$ to calculate the most confused class score as feedback...”
> “For the initial iteration, the random perturbation noise ($\mathcal{U}$) is added in the positive direction. The gradient (we modified it as noise) is added in the negative direction for further iterations and is changed to random perturbations (we modified it as a new random perturbation noise) in the subsequent iterations.”
>
> We will also revise the accompanying text in the manuscript to make these steps more explicit and eliminate ambiguity. We hope these changes address the concern.

---

> ### Author Response · Authors · 2025-06-13
> **3. Clarification on "1A" and Krum vs. No-defense GTA**
>
> Thank you for pointing out the ambiguity regarding the term "1A" in Table 9. We apologize for not defining it explicitly. As noted in Section 5 (page 12, paragraph 2, line 6), we evaluate F-CimBA under two attack configurations: single-client and multi-client. The notation "1A" in Table 9 refers to the single-client attacker setting, i.e., one adversarial client among all participants. To avoid confusion, we will ensure this is clearly defined in the revised manuscript.
>
> Regarding the observation that GTA is lower when the Krum defense is applied (Table 9: 73.95%) than when no defense is used (Table 6: 78.43%) for the GTSRB dataset: this is an insightful observation and we thank you for catching it.
>
> The reason lies in both the nature of the **Krum aggregation rule** and the **low attack visibility of F-CimBA's perturbations**. Krum is designed to improve robustness by selecting the client update that is closest (in Euclidean distance) to the majority of other client updates, thereby aiming to filter out anomalous or malicious ones. In our **single-client attack setting without Krum** (i.e., no defense, 1 out of 40 clients is adversarial), the global model is aggregated using all 40 client updates. In this case, the influence of the 39 benign clients naturally dilutes the effect of the poisoned update, resulting in a GTA of 78.43%. However, this still reflects a significant 11% degradation from the clean baseline (as noted in Table 5 and on page 13, paragraph 5, line 3), indicating that F-CimBA is effective without a defense in place. **When Krum is applied**, it attempts to select the most consistent (least deviant) update. Due to F-CimBA's design, which produces low-attack visibility and stealthy update, it can be stealthily considered benign and selected by Krum as the global model. This results in a more direct propagation of the poisoned update and, hence, a slightly lower GTA (73.95%).
>
> This outcome demonstrates two important aspects of our attack
>
> 1. F-CimBA exhibits low attack visibility, making it more difficult for defenses like Krum to detect and reject it.
>
> 2. Krum's selection strategy can be misled under our attack, which reduces its robustness in this scenario.
>
> We will incorporate this explanation into the main text to clearly account for the observed result and remove any confusion regarding its interpretation. We appreciate your attention to this important detail. We hope this clarifies the reasoning behind the observed results and fully addresses the question.

---

> ### Author Response · Authors · 2025-06-16
> **Response to the broader impact concerns**
>
> We thank you for raising this important concern. We fully agree that research involving adversarial attacks must be framed carefully to avoid potential misuse. However, the primary intent of our work is not to promote harmful use but rather to advance the understanding of vulnerabilities in FL systems, with the goal of improving their resilience and trustworthiness.
>
> Our work is aligned with a well-established line of research in FL security that emphasizes the importance of analyzing and exposing weaknesses in order to design better defenses, as seen in the following:
>
> * Shejwalkar et al. [1] comprehensively evaluate untargeted poisoning attacks and confirm that malicious updates can successfully bias the global model when not effectively filtered by the defense, even under constrained adversarial capabilities.
>
> * Usynin et al. [2] highlight that adversarial interference in collaborative learning leads to meaningful reductions in utility, even when perturbations are subtle. They show that even privacy-preserving FL systems are vulnerable to low-visibility attacks that do not require gradient access yet can significantly undermine the utility of the global model over multiple communication rounds.
>
> * Kumar et al. [3] explain that attacks with low attack visibility and small budgets can still have a high impact, particularly when the global model incorporates updates without sufficient scrutiny. These insights reinforce that well-designed poisoning attacks do not need to be large in magnitude to be impactful.
>
> Similarly, F-CimBA is proposed as a realistic, black-box, low-visibility threat model to fill the gap between simplistic adversarial settings and production-ready FL deployments. By rigorously studying how such attacks can succeed, despite restricted attacker capabilities, we aim to proactively inform the design of more effective defense strategies, including those evaluated in our work (e.g., Krum, LoMar, FLDefender).
>
> We have also ensured that the methodology is described from a scientific and diagnostic standpoint, avoiding weaponization or any recommendations for misuse. We will further revise the Broader Impact Statement in the manuscript to explicitly clarify that our work intends to strengthen FL systems by highlighting their robustness under practical threat scenarios. We hope this addresses the concern.
>
> [1] Shejwalkar et al. "**Back to the drawing board: A critical evaluation of poisoning attacks on production federated learning**," in *IEEE S&P*, 2022.
>
> [2] Usynin et al. "**Adversarial interference and its mitigations in privacy-preserving collaborative machine learning,**" in *Nature Machine Intelligence*, 2021.
>
> [3] Kumar et al. "**The impact of adversarial attacks on federated learning: A survey,**" in *IEEE TPAMI*, 2024.

---

### Decision · Action_Editor_xYyJ · 2025-08-05

**Recommendation:** Reject

**Additional Comments:**

The authors may consider submitting a major revision at a later time once they have carefully implemented and double-checked the changes sketched in their responses, in particular the new convergence result along with its proof and associated assumptions.

**Audience:**

Yes

**Audience Explanation:**

Robust Federated Learning is of broad interest to the TMLR community

**Claims And Evidence:**

No

**Claims Explanation:**

One reviewer identified an error in the convergence proof. During the discussion phase, the authors proposed a correction, which also turned out to be incorrect. They have now presented a third version of the proof, which introduces additional assumptions and alters the final result. Even if this new proof is correct, the changes to the paper are substantial and, in my view, warrant a new evaluation.

**Resubmission Of Major Revision:**

The authors may consider submitting a major revision at a later time.